# THE LESS YOU DEPEND, THE MORE YOU LEARN: SYNTHESIZING NOVEL VIEWS FROM SPARSE, UNPOSED IMAGES WITH MINIMAL 3D KNOWLEDGE

**Haoru Wang**[*]
Peking University
ou524u@stu.pku.edu.cn

**Kai Ye**[*]
Peking University
ye_kai@pku.edu.cn

**Minghan Qin**
ByteDance Seed
minghan@bytedance.com

**Yangyan Li**[†]
Ant Group
yangyan.lyy@antgroup.com

**Wenzheng Chen**[†]
Peking University & Beijing
Academy of Artificial Intelligence
wenzhengchen@pku.edu.cn

**Baoquan Chen**[†]
Peking University
baoquan@pku.edu.cn

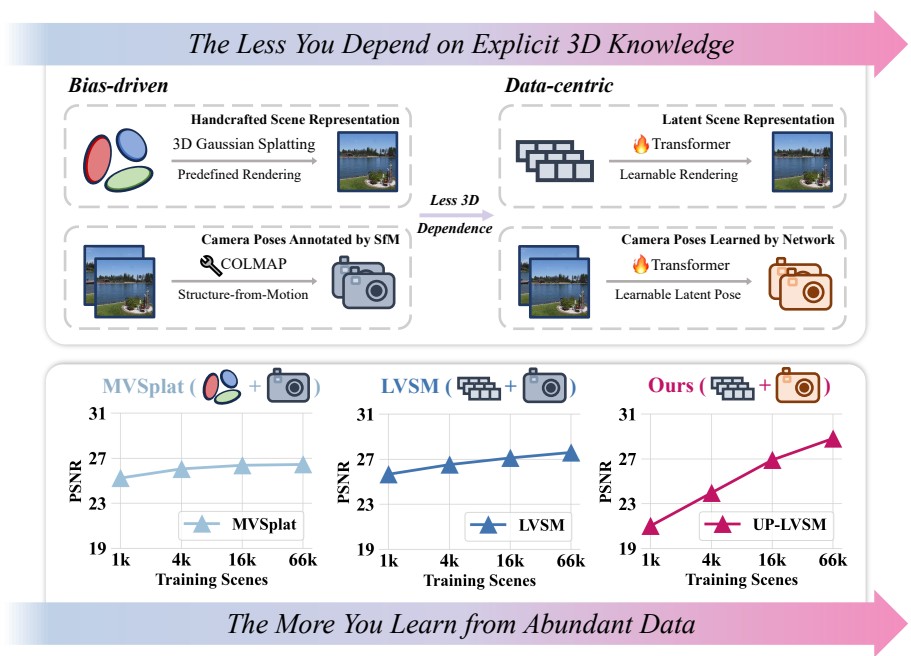

## ABSTRACT

Recent advances in feed-forward Novel View Synthesis (NVS) have led to a divergence between two design philosophies: *bias-driven methods*, which rely on explicit 3D knowledge, such as handcrafted 3D representations (*e.g.*, NeRF and 3DGS) and camera poses annotated by Structure-from-Motion algorithms, and *data-centric methods*, which learn to understand 3D structure implicitly from large-scale imagery data. This raises a fundamental question: which paradigm is more scalable in an era of ever-increasing data availability? In this work, we conduct a comprehensive analysis of existing methods and uncover a critical trend that the performance of methods requiring less 3D knowledge accelerates more as training data increases, eventually outperforming their 3D knowledge-driven counterparts, which we term *"the less you depend, the more you learn."* Guided by this finding, we design a feed-forward NVS framework that removes both explicit scene structure and pose annotation reliance. By eliminating these dependencies, our method leverages great scalability, learning implicit 3D awareness directly from vast quantities of 2D images, without any pose information for training or inference. Extensive experiments demonstrate that our model achieves state-of-the-art NVS performance, even outperforming methods relying on posed training data. The

---

[*]Equal contribution. [†]Equal advisory.

results validate not only the effectiveness of our data-centric paradigm but also the power of our scalability finding as a guiding principle.

# 1 INTRODUCTION

Novel View Synthesis (NVS), a long-standing challenge in computer vision and graphics, aims to render high-fidelity, unseen views of a scene from a collection of 2D images. Traditional solution typically involves Structure-from-Motion (SfM) (Wu et al., 2011; Schonberger & Frahm, 2016) to estimate camera parameters of each view, followed by per-scene fitting of representations like Neural Radiance Fields (Mildenhall et al., 2020, NeRF) or 3D Gaussian Splatting (Kerbl et al., 2023, 3DGS). Recently, the field has experienced a rapid shift towards a feed-forward paradigm (Yu et al., 2021; Charatan et al., 2024; Chen et al., 2024; Ye et al., 2025; Jin et al., 2025; Jiang et al., 2025), where the specific scene representations are directly predicted by neural networks instead of by a gradient-descent-based optimization. Leveraging powerful priors learned from large-scale datasets, these approaches can synthesize compelling novel views from sparse, wide-baseline, or even entirely unposed input images, bypassing the restrictive assumptions of their optimization-based predecessors.

Reviewing recent advances in feed-forward NVS reveals a key distinction between two design philosophies. The *bias-driven* one (Yu et al., 2021; Charatan et al., 2024; Chen et al., 2024; Ye et al., 2025) explicitly injects *3D knowledge*, such as human inductive biases (*e.g.*, handcrafted rendering formulas and predefined 3D representations) or intermediate 3D clues estimated by heuristic algorithms (*e.g.*, camera parameters obtained by COLMAP (Schonberger & Frahm, 2016))—directly into the method architecture. The alternative, the *data-centric* approaches (Sajjadi et al., 2023; Jin et al., 2025; Jiang et al., 2025), seek to learn *3D knowledge* implicitly, allowing spatial understanding to be distilled directly from vast quantities of 2D image data. This divergence raises fundamental questions about the future of the field: which paradigm proves more effective and scalable, especially in an era of increasingly abundant data?

In this work, we investigate the relationship between explicit *3D knowledge* dependencies and data scalability to address these questions. We categorize existing methods (Charatan et al., 2024; Chen et al., 2024; Ye et al., 2025; Jin et al., 2025) by their dependence on 3D knowledge and systematically analyze their performance across varying data regimes. Our experiments reveal a consistent and critical trend: methods that require less explicit 3D knowledge demonstrate superior data scalability. Their performance accelerates more significantly as the amount of training data increases, eventually surpassing their 3D knowledge-driven counterparts. This finding highlights a fundamental trade-off: while explicit 3D knowledge provides a useful scaffold for training on limited data, it creates a performance bottleneck at scale. We conclude that reducing dependence on 3D knowledge is essential for developing truly scalable NVS approaches.

Building on these insights, we propose UP-LVSM, a data-centric NVS framework designed to unlock scalability by eliminating 3D knowledge dependencies. Using a pure Transformer architecture similar to Jin et al. (2025), UP-LVSM models scenes implicitly within a latent space, bypassing the need for predefined 3D structures. We further identify camera poses annotated by Structure-from-Motion pipelines as an indirect form of 3D knowledge that hinders scalability. To address this, we introduce a novel *Latent Plücker Learner* to infer camera geometry directly from images in a self-supervised manner, further bypassing the need for pose annotations during training.

By shedding these dependencies on 3D knowledge, UP-LVSM fully leverages data scaling to synthesize photorealistic and 3D-consistent novel views from sparse, unposed images—even without any pose supervision during training. Experiments demonstrate that UP-LVSM outperforms state-of-the-art approaches that rely on explicit scene structure or pose annotations. This not only confirms the viability of minimizing 3D knowledge but also establishes a new path toward scalable and generalizable novel view synthesis learned purely from 2D observations.

Our primary contributions are summarized as follows:

1. We perform a systematic analysis of NVS methods through the lens of 3D knowledge, uncovering the key principle that reducing dependence on such knowledge is the key to unlocking scalability.

2. We propose UP-LVSM, a novel data-centric NVS framework that effectively learns spatial reasoning from unposed 2D images without requiring explicit 3D representations or pose supervision.

Extensive experiments demonstrate that our framework achieves both superior scalability and state-of-the-art performance, validating our key hypothesis and the effectiveness of the data-centric paradigm.

## 2 REVISITING 3D KNOWLEDGE IN FEED-FORWARD NOVEL VIEW SYNTHESIS

### 2.1 PRELIMINARIES

**Novel View Synthesis** The goal of novel view synthesis (NVS) is to reconstruct a 3D scene representation, denoted as $\mathcal{S}$, from a given set of 2D images and their corresponding camera poses, $\{(\mathcal{I}^i, \mathcal{P}_\mathcal{I}^i)\}_{i=1}^N$. This reconstructed scene is then used to render a novel image $\mathcal{T}$ from a new target viewpoint $\mathcal{P}_\mathcal{T}$. The process is typically supervised by a reconstruction loss $\mathcal{L}(\mathcal{T}, \tilde{\mathcal{T}})$ that measures the difference between the rendered image $\mathcal{T}$ and the ground-truth image $\tilde{\mathcal{T}}$. This can be formally expressed as:

$$\mathcal{S} = \mathcal{A}(\mathcal{I}^1, \mathcal{P}_\mathcal{I}^1, \mathcal{I}^2, \mathcal{P}_\mathcal{I}^2, ..., \mathcal{I}^N, \mathcal{P}_\mathcal{I}^N), \quad \mathcal{T} = \mathcal{R}(\mathcal{S}, \mathcal{P}_\mathcal{T}), \tag{1}$$

where $\mathcal{A}$ is the scene reconstruction function and $\mathcal{R}$ is the rendering function. In settings with dense observations ($N \geq 50$), the scene $\mathcal{S}$ is typically modeled using explicit 3D representations like Neural Radiance Fields (NeRFs) (Mildenhall et al., 2020) or 3D Gaussian Splatting (3DGS) (Kerbl et al., 2023). The differentiable nature of these representations allows the reconstruction function $\mathcal{A}$ to be implemented as a per-scene optimization process (Mildenhall et al., 2020; Barron et al., 2022; Kerbl et al., 2023; Yu et al., 2024), which effectively yields photorealistic novel views.

**Feed-Forward NVS** Despite their promise, per-scene optimization approaches are limited by their reliance on dense observations, making them less suitable for under-constrained settings where inputs are sparse (typically $N \leq 5$) or camera poses $\mathcal{P}_\mathcal{I}$ are unavailable. To address this, feed-forward approaches (Yu et al., 2021; Charatan et al., 2024; Chen et al., 2024; Ye et al., 2025; Jin et al., 2025) employ a neural network as the reconstruction function $\mathcal{A}$. By training on large-scale multi-view datasets (Zhou et al., 2018; Deitke et al., 2023; Ling et al., 2024), these methods learn powerful priors to compensate for the ambiguity of sparse inputs, enabling reconstruction of the scene $\mathcal{S}$ in a single forward pass. More discussions are provided in Appendix B.

### 2.2 3D KNOWLEDGE DEPENDENCE IN FEED-FORWARD NVS

In reviewing recent advances in feed-forward NVS, the methods can be characterized by their varying reliance on 3D knowledge, which typically manifests in two key aspects: explicit scene structure and pose annotation availability. Table 1 demonstrates the categorization.

| Method | Explicit Scene Structure | $\mathcal{S}$ Modeling | $\mathcal{R}$ Modeling | Problem Setting | $\mathcal{P}_\mathcal{I}$ | $\mathcal{P}_\mathcal{T}$ |
|---|---|---|---|---|---|---|
| PixelNeRF (Yu et al., 2021) | ✓ | NeRF | Volumetric Rendering | posed | ✓ | ✓ |
| PixelSplat (Charatan et al., 2024) | ✓ | 3DGS | Gaussian Splatting | | ✓ | ✓ |
| MVSplat (Chen et al., 2024) | ✓ | 3DGS | Gaussian Splatting | | ✓ | ✓ |
| LVSM (Jin et al., 2025) | ✗ | Latent | Learnable Network | | ✓ | ✓ |
| NoPoSplat (Ye et al., 2025) | ✓ | 3DGS | Gaussian Splatting | posed-target | ✗ | ✓ |
| Ours (PT-LVSM) | ✗ | Latent | Learnable Network | | ✗ | ✓ |
| SPFSplat* (Huang & Mikolajczyk, 2025) | ✓ | 3DGS | Gaussian Splatting | unposed | ✗ | ✗ |
| Rayzer* (Jiang et al., 2025) | ✗ | Latent | Learnable Network | | ✗ | ✗ |
| Ours (UP-LVSM) | ✗ | Latent | Learnable Network | | ✗ | ✗ |

Table 1: **3D Knowledge in Feed-Forward NVS.** We characterize recent feed-forward NVS methods (*denotes concurrent work) based on their varying dependence on explicit 3D knowledge (*i.e.*, the choice of $\mathcal{S}$ and $\mathcal{R}$ modeling) and pose availability (*i.e.*, whether $\mathcal{P}_\mathcal{I}$ and $\mathcal{P}_\mathcal{T}$ are provided).

**Explicit Scene Structure** As demonstrated in Table 1, this refers to the integration of explicit 3D representations or handcrafted rendering operations directly into the NVS architecture. Methods based on established 3D structures, such as PixelNeRF (Yu et al., 2021), PixelSplat (Charatan et al., 2024), MVSplat (Chen et al., 2024), NoPoSplat (Ye et al., 2025), and SPFSplat (Huang & Mikolajczyk, 2025), incorporate explicit representations like NeRF (Mildenhall et al., 2020) or 3DGS (Kerbl et al., 2023) along with their associated rendering equations to model the scene $\mathcal{S}$ and the render function $\mathcal{R}$. These architectural choices enforce a strong geometric consistency based on principles like volumetric rendering or plane sweeps, thereby explicitly injecting 3D knowledge into method designs. In contrast, approaches like LVSM (Jin et al., 2025) and Rayzer (Jiang et al., 2025), treat scene modeling as a learning problem, representing the scene $\mathcal{S}$ implicitly as latent tokens and allowing its modeling to be learned implicitly from data.

**Pose Annotation Availability** Another key aspect is the availability of camera pose annotations for input views $\mathcal{P}_{\mathcal{I}}$ and target views $\mathcal{P}_{\mathcal{T}}$. This dependency defines the problem's constraints, leading to three primary settings, as detailed in Figure 1 and Table 1: (1) **posed**: camera poses are required for both input and target views. (2) **posed-target**: input view poses are unknown, but target view poses are required for supervision. (3) **unposed**: no camera pose information is assumed for either input or target views. Detailed problem setting definitions can be found in Appendix A. While the first two settings rely on datasets with poses often generated by SfM pipelines like COLMAP (Schonberger & Frahm, 2016), it is important to recognize that SfM itself is built on heuristic algorithms and geometric inductive biases and often produces incorrect estimations. Consequently, relying on these poses is an indirect form of dependence on 3D knowledge. In this context, only the *unposed* setting truly operates without such dependence, typically relying on a latent pose learned in a self-supervised manner (Sajjadi et al., 2023).

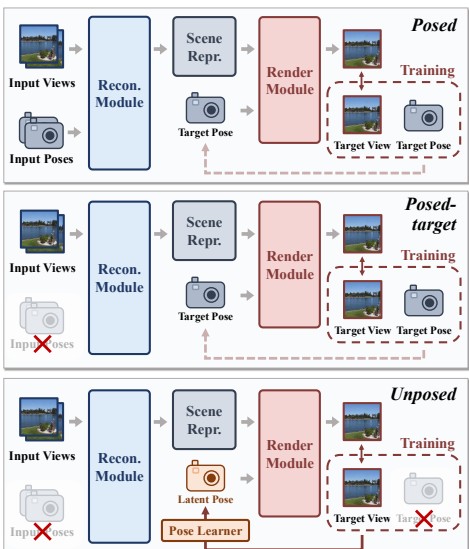

Figure 1: Pose Annotation Availability.

## 2.3 EXPLICIT VS. LEARNED 3D KNOWLEDGE: A DIVERGENCE

Viewing predefined 3D structure and pose availability as forms of explicit 3D knowledge reveals a fundamental divergence in feed-forward NVS design. On one side, the bias-driven paradigm relies on explicitly injecting 3D knowledge as human inductive biases. On the other, the data-centric paradigm allows this knowledge to be learned implicitly from large-scale imagery. While both paradigms have proven effective, a critical question remains: which is more scalable and learns more effectively in an era of increasing data abundance? This work argues that reducing dependence on explicit 3D knowledge leads to superior scalability and, ultimately, better performance. We term this principle *"the less you depend, the more you learn"*, and provide a detailed analysis to substantiate this hypothesis in the following section.

## 3 THE LESS YOU DEPEND, THE MORE YOU LEARN

In this section, we present our analysis to validate the hypothesis that *the less you depend, the more you learn*. Specifically, *the less you depend* refers to reducing reliance on 3D knowledge in methodology design, including explicit scene structure and camera pose annotations. Meanwhile, *the more you learn* refers to scalability, which is defined as how performance improves as the amount of training data increases. By examining the relationship between performance and data quantity for different methods, we find the performance of methods that requires less 3D knowledge accelerates more as data scales.

**Dataset** While our experiments are conducted across diverse datasets (Zhou et al., 2018; Ling et al., 2024; Liu et al., 2021; Deitke et al., 2023), as explained later in Section 3.3, we select the RealEstate10K dataset (Zhou et al., 2018) as the representative to introduce our experimental setup, which contains real-world imagery over 70K scenes. To evaluate scalability, we create four training subsets of increasing size (little, medium, large, and full, summarized in Table 2), while using a single, consistent test set for all evaluations to ensure fair comparison.

| Subset | Train | Test |
|--------|-------|------|
| Little | 1202 | |
| Medium | 4121 | 7286 |
| Large | 16449 | |
| Full | 66033 | |

Table 2: Number of scenes in RealEstate10K subsets.

**Representative Methods** We select methods that span different levels of scene structural biases and vary in problem settings to represent different dependencies on 3D knowledge. In the *posed* setting, we contrast the structural bias-driven MVSplat (Chen et al., 2024) with the bias-free LVSM (Jin et al., 2025). In the *posed-target* setting, we select NoPoSplat (Ye et al., 2025) as the bias-driven representative. As no established bias-free method exists in the *posed-target* setting, we simply

adapt LVSM for this setting to further enhance the comprehensiveness of our analysis, denoting it as PT-LVSM. We leave adaptation details in Appendix I to maintain the flow of our main analysis.

**Experimental Results**   We evaluate these methods trained at different subsets of the RealEstate10K dataset to assess their scalability, as shown in Table 3. We quantify scalability as the average gain in NVS metrics (PSNR, SSIM, LPIPS) for every $4\times$ increase in training data.

| Method | Bias-free | $\mathcal{P}_{\mathcal{I}}$-free | *Little* Subset | *Medium* Subset
PSNR↑ / SSIM↑ / LPIPS↓ | *Large* Subset | *Full* Subset | Avg. Gain ↑
ΔPSNR / ΔSSIM / ΔLPIPS |
|---|---|---|---|---|---|---|---|
| MVSplat | ✗ | ✗ | 25.24 / 0.849 / 0.136 | 26.06 / 0.865 / 0.128 | 26.38 / 0.872 / 0.124 | 26.45 / 0.874 / 0.123 | 0.39 / 0.008 / 0.004 |
| LVSM | ✓ | ✗ | 25.67 / 0.831 / 0.145 | 26.52 / 0.851 / 0.135 | 27.11 / 0.864 / 0.124 | 27.60 / 0.874 / 0.117 | 0.64 / 0.014 / 0.009 |
| NoPoSplat | ✗ | ✓ | 25.09 / 0.840 / 0.142 | 25.33 / 0.849 / 0.139 | 25.43 / 0.851 / 0.139 | 25.46 / 0.854 / 0.137 | 0.12 / 0.004 / 0.002 |
| PT-LVSM | ✓ | ✓ | 20.80 / 0.659 / 0.231 | 22.92 / 0.731 / 0.184 | 24.54 / 0.781 / 0.159 | 26.00 / 0.825 / 0.135 | **1.72 / 0.055 / 0.031** |

Table 3: Scalability Comparisons on RealEstate10K (Zhou et al., 2018).

## 3.1 DEPENDENCY ANALYSIS ON EXPLICIT SCENE STRUCTURE

As shown in Table 3 and Figure 2, our experiments reveal a clear trade-off between explicit scene structure and data scalability. Methods with explicit scene structure (*i.e.*, MVSplat and NoPoSplat) excel in low-data regimes (*e.g.*, 1K scenes) but fail to scale effectively with more data. Conversely, implicit methods (*i.e.*, LVSM and PT-LVSM), while initially underperforming, demonstrate substantial performance gains as the training set grows to 66K scenes. This confirms our hypothesis that reducing reliance on explicit 3D structures is crucial for unlocking data scalability.

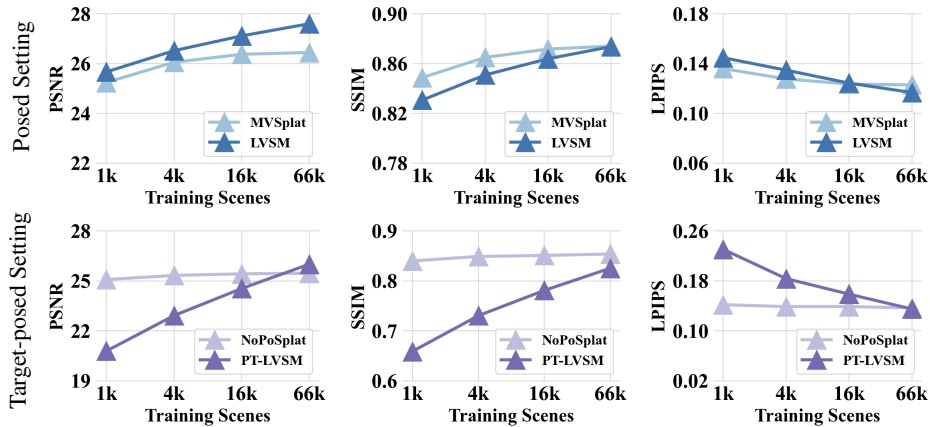

Figure 2: Scalability Comparison on Varying Choices of Scene Structure.

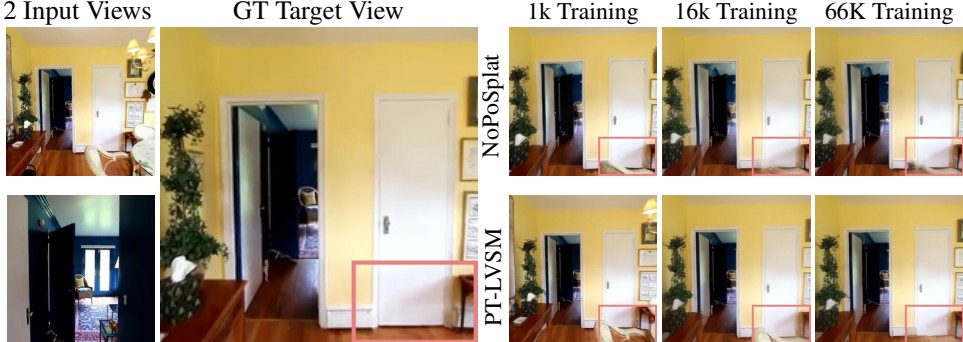

Figure 3: **Intuitive Explanation.** On posed-target setting, both NoPoSplat and PT-LVSM fail to infer correct spatial structure when trained with 1K scenes, resulting in artifacts at the right bottom of target views. While bias-driven NoPoSplat consistently makes mistakes, PT-LVSM significantly improves when training data scales up from 1K to 66K, eventually outperforming NoPoSplat.

**Discussion**   Intuitively, strong structural biases act as a necessary scaffold when training data is scarce, compensating for a lack of information. However, as data becomes abundant, these same biases become restrictive, limiting the model's ability to learn complex patterns directly from the data and thus hindering generalization. As illustrated in Figure 3, performance of the data-centric method improves with more data, while the bias-driven approach stagnates.

## 3.2 DEPENDENCY ANALYSIS ON POSE ANNOTATIONS

Beyond 3D bias, we find that reliance on pose annotations also critically impacts scalability. As shown in Table 3 and Figures 4, PT-LVSM demonstrates significantly better scalability than LVSM. Although both are data-centric methods with weak 3D biases, they differ in a key aspect: LVSM requires input images annotated with camera pose $\mathcal{P}_{\mathcal{I}}$, whereas PT-LVSM does not.

**Discussion** Benefiting from known camera poses that provide strong 3D clues, posed methods like LVSM should theoretically have a significantly higher performance ceiling. However, it is observed that our pose-free PT-LVSM quickly closes this gap as data scales. We attribute this to noise in pose annotations. As discussed in Section 2.2, pose annotations in real-world datasets (Zhou et al., 2018; Yao et al., 2020; Yeshwanth et al., 2023; Ling et al., 2024) are typically generated by Structure-from-Motion tools (Wu et al., 2011; Schonberger & Frahm, 2016) that are built on geometric inductive biases, thereby introducing noise and inconsistencies.

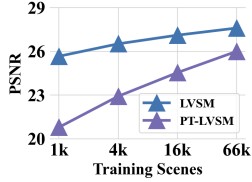

Figure 4: Scalability of LVSM and PT-LVSM.

We argue that relying on these poses during training is an indirect form of 3D knowledge dependence, which creates a bottleneck at scale. See Appendix E for a detailed explanation.

## 3.3 UNLOCKING DATA-CENTRIC FEED-FORWARD NVS

**Motivation** Our analysis indicates that data scalability in feed-forward NVS is fundamentally limited by dependencies on explicit scene structure and camera pose annotations. As PT-LVSM still relies on target view poses $\mathcal{P}_{\mathcal{T}}$ for its posed-target setting (Figure 1), a critical question remains: can we achieve even greater scalability and surpass the performance ceiling of pose-dependent methods like LVSM by removing this final dependency? We answer this by proposing **UP-LVSM**, a novel feed-forward NVS framework that learns 3D knowledge implicitly from 2D images without any pose annotations, which we will detail in Section 4.

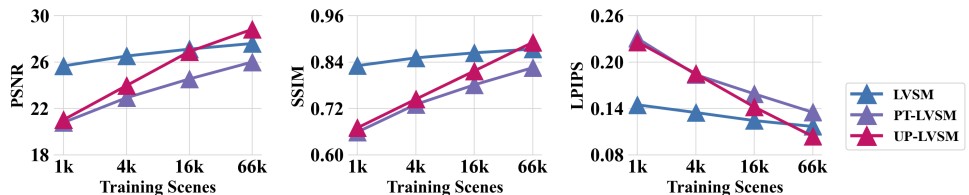

Figure 5: Superior Scalability of UP-LVSM on RealEstate10K.

| Method | $\mathcal{P}_{\mathcal{I}}$-free | $\mathcal{P}_{\mathcal{T}}$-free | RealEstate10K (Zhou et al., 2018): PSNR↑ / SSIM↑ / LPIPS↓ | | | | ΔPSNR / ΔSSIM / ΔLPIPS |
|---|---|---|---|---|---|---|---|
| | | | #Scenes: 1K | #Scenes: 4K | #Scenes: 16K | #Scenes: 66K | Avg. Gain ↑ |
| LVSM | ✗ | ✗ | 25.67 / 0.831 / 0.145 | 26.52 / 0.851 / 0.135 | 27.11 / 0.864 / 0.124 | 27.60 / 0.874 / 0.117 | 0.64 / 0.014 / 0.009 |
| PT-LVSM | ✓ | ✗ | 20.80 / 0.659 / 0.231 | 22.92 / 0.731 / 0.184 | 24.54 / 0.781 / 0.159 | 26.00 / 0.825 / 0.135 | 1.72 / 0.055 / 0.031 |
| UP-LVSM | ✓ | ✓ | 21.03 / 0.670 / 0.226 | 23.97 / 0.744 / 0.185 | 26.90 / 0.817 / 0.142 | 28.82 / 0.891 / 0.104 | **2.63 / 0.074 / 0.041** |

| Method | $\mathcal{P}_{\mathcal{I}}$-free | $\mathcal{P}_{\mathcal{T}}$-free | DL3DV (Ling et al., 2024): PSNR↑ / SSIM↑ / LPIPS↓ | | | | ΔPSNR / ΔSSIM / ΔLPIPS |
|---|---|---|---|---|---|---|---|
| | | | #Scenes: 0.2K | #Scenes: 0.6K | #Scenes: 2.5K | #Scenes: 10K | Avg. Gain ↑ |
| LVSM | ✗ | ✗ | Not Converged. | 16.61 / 0.531 / 0.457 | 17.82 / 0.572 / 0.422 | 19.14 / 0.603 / 0.397 | 1.27 / 0.035 / 0.030 |
| PT-LVSM | ✓ | ✗ | | 16.58 / 0.410 / 0.465 | 17.66 / 0.585 / 0.424 | 19.47 / 0.641 / 0.379 | 1.44 / 0.115 / 0.043 |
| UP-LVSM | ✓ | ✓ | | 16.45 / 0.387 / 0.471 | 17.66 / 0.581 / 0.423 | 19.59 / 0.653 / 0.366 | **1.57 / 0.133 / 0.053** |

| Method | $\mathcal{P}_{\mathcal{I}}$-free | $\mathcal{P}_{\mathcal{T}}$-free | ACID (Liu et al., 2021): PSNR↑ / SSIM↑ / LPIPS↓ | | | | ΔPSNR / ΔSSIM / ΔLPIPS |
|---|---|---|---|---|---|---|---|
| | | | #Scenes: 0.2K | #Scenes: 0.8K | #Scenes: 3K | #Scenes: 13K | Avg. Gain ↑ |
| LVSM | ✗ | ✗ | Not Converged. | 23.43 / 0.717 / 0.245 | 25.96 / 0.759 / 0.223 | 27.01 / 0.779 / 0.211 | 1.79 / 0.031 / 0.017 |
| PT-LVSM | ✓ | ✗ | | 18.41 / 0.565 / 0.459 | 26.31 / 0.760 / 0.204 | 26.75 / 0.768 / 0.199 | 4.17 / 0.102 / 0.130 |
| UP-LVSM | ✓ | ✓ | | 15.92 / 0.431 / 0.643 | 26.88 / 0.771 / 0.194 | 27.21 / 0.787 / 0.186 | **5.65 / 0.178 / 0.224** |

| Method | $\mathcal{P}_{\mathcal{I}}$-free | $\mathcal{P}_{\mathcal{T}}$-free | Objaverse (Deitke et al., 2023): PSNR↑ / SSIM↑ / LPIPS↓ | | | | ΔPSNR / ΔSSIM / ΔLPIPS |
|---|---|---|---|---|---|---|---|
| | | | #Objects: 2K | #Objects: 8K | #Objects: 32K | #Objects: 128K | Avg. Gain ↑ |
| LVSM | ✗ | ✗ | 24.58 / 0.814 / 0.177 | 28.90 / 0.887 / 0.106 | 29.63 / 0.898 / 0.096 | 30.22 / 0.906 / 0.087 | 1.77 / 0.029 / 0.028 |
| PT-LVSM | ✓ | ✗ | 21.64 / 0.754 / 0.326 | 25.01 / 0.825 / 0.198 | 26.83 / 0.852 / 0.141 | 27.44 / 0.859 / 0.120 | 1.92 / 0.034 / 0.068 |
| UP-LVSM | ✓ | ✓ | 19.96 / 0.712 / 0.403 | 23.38 / 0.773 / 0.275 | 26.02 / 0.827 / 0.158 | 26.12 / 0.829 / 0.156 | **2.11 / 0.040 / 0.086** |

Table 4: We conduct extensive experiments across diverse datasets (Zhou et al., 2018; Ling et al., 2024; Liu et al., 2021; Deitke et al., 2023) to validate our hypothesis.

**Discussion** As demonstrated in Figure 5 and Table 4, our UP-LVSM achieves consistently superior scalability by fully eliminating reliance on explicit 3D knowledge, which validates our hypothesis. Moreover, benefiting from great scalability, our UP-LVSM achieves state-of-the-art performance using only 2D supervision, which unlocks the full potential of data-driven NVS learning. More analysis about data scalability of our method is provided in Appendix E.

## 4 METHODOLOGY

As motivated in Section 3.3, we propose **UP-LVSM** (**Un**posed **L**arge **V**iew **S**ynthesis **M**odel) to unlock scalability by eliminating the need for explicit scene structure and camera pose annotations. This, however, places our method in the challenging *unposed* setting (Figure 1), where the core difficulty lies in learning without the explicit target pose supervision ($\mathcal{P}_\mathcal{T}$) available in simpler settings like posed-target. To this end, we propose the *Latent Plücker Learner*, the core component in UP-LVSM that learns a meaningful latent pose space in a self-supervised manner. In this section, we will detail these technical designs and provide experimental results to validate the effectiveness of our proposed method, highlighting its capability of synthesizing high-fidelity novel views directly from unstructured 2D image collections.

### 4.1 TRANSFORMER-BASED ARCHITECTURE

As illustrated in Figure 6 (a), we build upon previous works (Jin et al., 2025; Wang et al., 2025), employing Transformer (Vaswani et al., 2017) to construct our feed-forward neural networks as an encoder-decoder architecture. As detailed in Figure 6 (b), it first encodes input images $\mathcal{I}$ into patchified tokens using DINOv2 (Oquab et al., 2023), and then employs Transformer networks to reconstruct scene latents. A decoder takes the scene latents and the camera pose information in latent Plücker as inputs to synthesize novel views. See Appendix I for detailed network architecture.

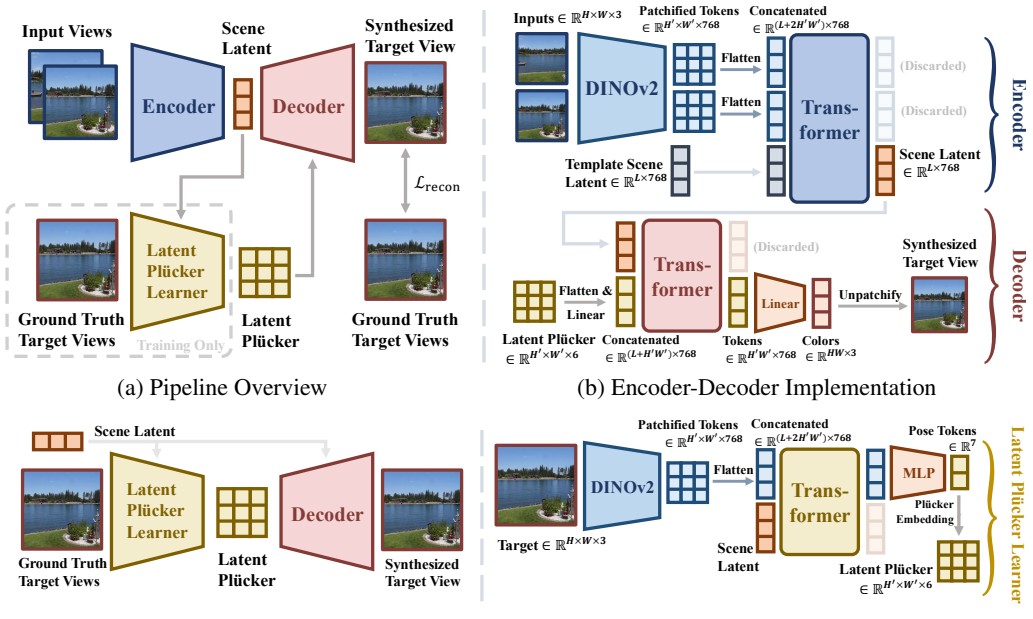

(a) Pipeline Overview          (b) Encoder-Decoder Implementation

(c) Autoencoder-like Latent Plücker Learning          (d) Latent Plücker Learner Implementation

Figure 6: UP-LVSM Architecture.

### 4.2 LATENT PLÜCKER LEARNER

**Background**   As previously discussed, UP-LVSM operates in the challenging unposed setting, where ground-truth poses $\mathcal{P}_\mathcal{T}$ are not provided. Target poses are fundamental to conventional NVS (*posed & posed-target* settings), as they provide the explicit viewpoint conditioning required for the rendering process. This creates clear image-pose pairs for supervised learning. In the absence of such ground-truth, the model must learn to infer latent poses in a self-supervised manner, leveraging the implicit signal from multi-view imagery (*i.e.*, images of the same scene serve as positive pairs). However, the key challenge lies in constraining the representational capacity of this learned latent pose. A high-dimensional latent space risks severe information leakage, where the latent pose inadvertently encodes the target image itself rather than just the viewpoint. Conversely, a low-dimensional space may lack the expressiveness required to guide fine-grained, pixel-accurate rendering.

**Method**   We propose the *Latent Plücker Learner* to address this challenge using an autoencoder architecture that strategically manages information flow, as illustrated in Figure 6 (c). To prevent

information leakage from the target view, the learner first uses an encoder to distill the image into a highly compact 7D latent pose token (translation $\mathbf{x}$ and quaternion $\mathbf{q}$), as detailed in Figure 6 (d). This low-dimensional bottleneck constrains the latent space, making it unable to retain specific image content. Conversely, to ensure this compact representation is expressive enough for rendering, the token is then analytically upsampled into fine-grained, pixel-level conditions by adapting the Plücker ray embedding (Plucker, 1865) to operate within a learned latent space. This design provides rich, per-ray conditioning for the renderer while maintaining a minimal set of learnable pose parameters. By training this module with a shared latent space across scenes, our model effectively learns a meaningful camera pose representation without any 3D supervision.

**Plücker Ray Embedding**   Plücker ray embedding (Plucker, 1865) is an effective technique to embed camera pose information into pixel-aligned tokens. Given an image $\mathcal{I} \in \mathbb{R}^{H \times W \times 3}$, the Plücker ray encodes its corresponding camera pose for each pixel as $\hat{\mathcal{P}} = \text{concat}(\mathbf{o} \times \mathbf{d}, \mathbf{d}) \in \mathbb{R}^{H \times W \times 6}$, where $\mathbf{o}$ represents the camera center and $\mathbf{d}$ is the camera ray direction corresponding to the pixels.

## 4.3 EXPERIMENTS

**Setup**   As our method employs the DINOv2 tokenizer, the $256 \times 256$ resolution is incompatible with for a patch size of 14. We therefore adopt the $224 \times 224$ resolution to align with DINOv2's native configuration. Note that to ensure strict fairness, we do not use official checkpoints of baselines. Instead, we retrain all baseline methods from scratch using the exact same $224 \times 224$ resolution with a patch size of 14 and the same training split as our UP-LVSM. All training follows the official implementations of each baseline. This experimental setup is also applied in the scalability experiments of Section 3.

**NVS Performance**   We follow Ye et al. (2025) to evaluate NVS performance. Qualitative and quantitative results on the RealEstate10K dataset (Zhou et al., 2018) are shown in Figure 7 and Table 5, respectively. Despite trained without any 3D supervision, our UP-LVSM even outperforms previous pose-dependent methods, which have access to the dataset-provided poses. These results demonstrate the effectiveness of our proposed framework, validating the feasibility of scaling 2D-only learning frameworks to unlock spatial reasoning without explicit scene structure or pose annotations. More results are illustrated in Appendix F.

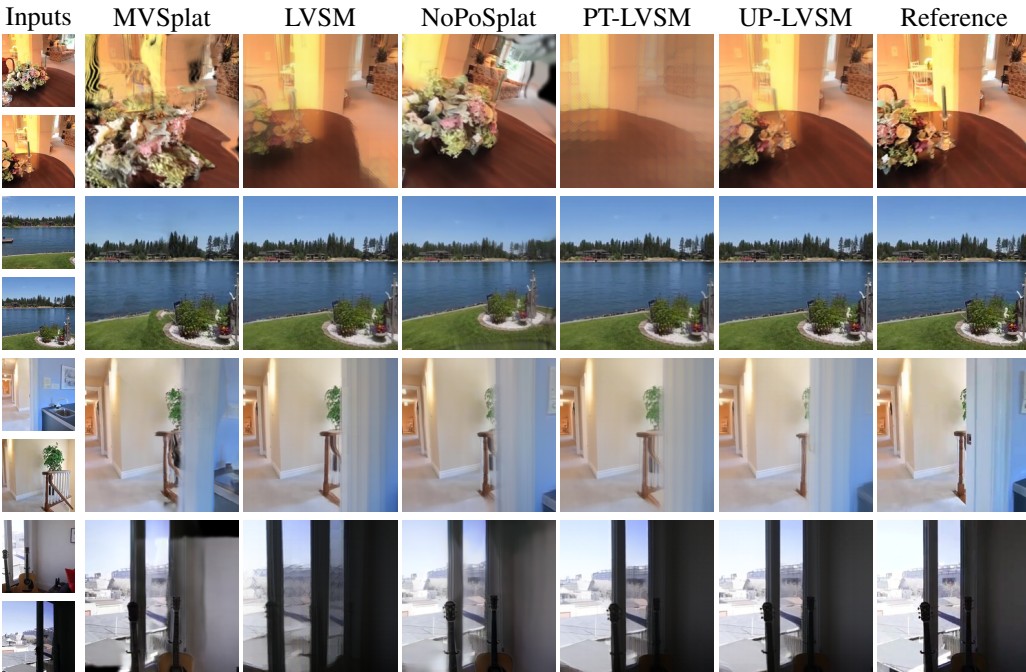

Figure 7: Qualitative View Synthesis Comparisons.

| Method | Input Pose $\mathcal{P}_\mathcal{I}$ | Large Overlap | | | Medium Overlap | | | Small Overlap | | | Full Eval. | | |
|---|---|---|---|---|---|---|---|---|---|---|---|---|---|
| | | PSNR ↑ | SSIM ↑ | LPIPS ↓ | PSNR ↑ | SSIM ↑ | LPIPS ↓ | PSNR ↑ | SSIM ↑ | LPIPS ↓ | PSNR ↑ | SSIM ↑ | LPIPS ↓ |
| PixelNeRF | | 20.94 | 0.581 | 0.517 | 20.38 | 0.559 | 0.540 | 19.27 | 0.536 | 0.568 | 20.33 | 0.572 | 0.549 |
| PixelSplat | ✓ | 26.18 | 0.879 | 0.115 | 23.61 | 0.821 | 0.162 | 21.22 | 0.752 | 0.225 | 25.51 | 0.867 | 0.126 |
| MVSplat | | 27.32 | 0.889 | 0.112 | 23.97 | 0.819 | 0.165 | 20.67 | 0.730 | 0.238 | 26.45 | 0.874 | 0.123 |
| LVSM | | 28.58 | 0.887 | 0.108 | 25.60 | 0.830 | 0.149 | 22.71 | 0.765 | 0.202 | 27.60 | 0.874 | 0.117 |
| NoPoSplat | | 25.84 | 0.854 | 0.133 | 23.67 | 0.808 | 0.177 | 21.58 | 0.750 | 0.231 | 25.46 | 0.854 | 0.137 |
| Ours (PT-LVSM) | ✗ | 26.47 | 0.829 | 0.130 | 24.27 | 0.778 | 0.173 | 22.03 | 0.720 | 0.224 | 26.00 | 0.825 | 0.135 |
| Ours (UP-LVSM) | | **29.51** | **0.901** | **0.098** | **26.93** | **0.852** | **0.132** | **24.54** | **0.796** | **0.174** | **28.82** | **0.891** | **0.104** |

Table 5: Quantitative Comparisons on RealEstate10K (Zhou et al., 2018). Following Ye et al. (2025), we conduct evaluations across different overlap levels. Our UP-LVSM outperforms pose-dependent approaches, particularly in challenging cases where input views share minimal overlap.

| Method | Train→Test | PSNR ↑ | SSIM ↑ | LPIPS ↓ |
|---|---|---|---|---|
| LVSM | ACID→ACID | 27.03 | 0.774 | 0.201 |
| UP-LVSM | ACID→ACID | 27.21 | 0.787 | 0.194 |
| UP-LVSM | RE10K→ACID | **27.33** | **0.792** | **0.182** |

Table 6: Zero-Shot Generalization.

| Model | Source of Target Pose $\mathcal{P}_\mathcal{T}$ | PSNR ↑ | SSIM ↑ | LPIPS ↓ |
|---|---|---|---|---|
| UP-LVSM | (a) SfM annotations | 26.00 | 0.825 | 0.135 |
| | (b) Pose Estimator in Sajjadi et al. (2023) | 20.92 | 0.521 | 0.558 |
| | (c) Latent Plücker Learner | **28.82** | **0.891** | **0.104** |

Table 7: Ablation Studies on the Latent Plücker Learner.

**Zero-Shot Generalization**    We evaluate the zero-shot generalization of UP-LVSM by training on the RealEstate10K dataset and testing on the unseen ACID dataset (Liu et al., 2021), denoted as RE10K→ACID. As demonstrated in Table 6, this zero-shot generalization even outperforms methods trained directly on ACID, highlighting the performance gains from the larger quantity of data in RealEstate10K (66K scenes) over ACID (13K scenes), which reinforces our findings.

**Ablation Studies**    To validate the effectiveness of our Latent Plücker Learner, we conduct ablation studies by contrasting different source of target pose $\mathcal{P}_\mathcal{T}$ information during UP-LVSM's training, as illustrated in Figure A. The choice of SfM annotations results in a fallback to the posed-target setting and the inaccuracies of SfM lead to a performance degradation, as demonstrated in Table 7 (a). We also replace the Latent Plücker Learner with the pose estimator from Sajjadi et al. (2023), which uses a key-value querying mechanism and masking strategy to encourage learning a meaningful latent pose space. While effective in smaller data regimes, this design becomes unstable at scale and yields suboptimal results, as demonstrated in Table 7 (b). In contrast, our Latent Plücker Learner leverages fine-grained Plücker embeddings together with a bottlenecked architecture, naturally avoiding information leakage and delivering substantially better rendering quality.

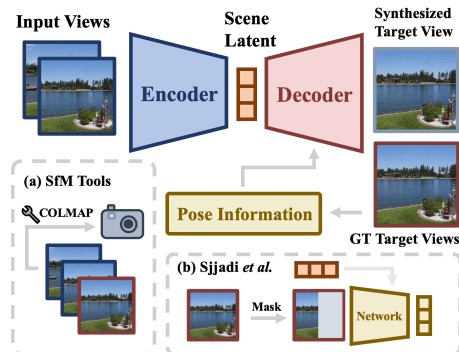

Figure A. **Ablation Studies on the Latent Plücker Learner.** (a) Directly using explicit poses annotated by SfM tools to provide pose information of target views. (b) Replacing the Latent Plücker Learner with a special pose estimator network (Sajjadi et al., 2023).

### 4.4 MORE INVESTIGATION

| Model | 0° ∼ 15° ↑ | 15° ∼ 30° ↑ | 30° ∼ 60° ↑ | 60° ∼ 180° ↑ |
|---|---|---|---|---|
| CLIP (Radford et al., 2021) | 6.6 | 5.2 | 4.7 | 3.0 |
| MAE (He et al., 2022) | 10.8 | 7.8 | 6.0 | 3.5 |
| DINOv2 (Oquab et al., 2023) | **36.8** | **27.5** | 17.9 | 8.0 |
| UP-LVSM (Ours) | 31.9 | 25.4 | **18.0** | **8.2** |

Table 8: Correspondence Estimation Accuracy for 3D Awareness Probing (El Banani et al., 2024).

| Model | PSNR@All ↑ | SSIM@All ↑ | LPIPS@All ↓ |
|---|---|---|---|
| DUSt3R (Wang et al., 2024) | **19.28** | **0.630** | **0.391** |
| LVSM (Jin et al., 2025) | 18.73 | 0.590 | 0.415 |
| UP-LVSM (Ours) | 18.81 | 0.601 | 0.409 |

Table 9: GTA Metrics for 3D Awareness Probing (Chen et al., 2025).

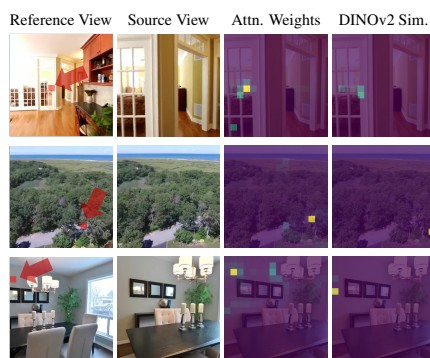

Figure 8: Attention Weight Visualization.

**Probing 3D Awareness**    As our UP-LVSM is a fully implicit approach to NVS which forgoes explicit 3D knowledge, it is critical to ascertain whether it implicitly learns spatial relationships. To this end, we probe the model's 3D awareness following the methodologies proposed in (El Banani

et al., 2024; Chen et al., 2025). Our quantitative assessment, summarized in Table 8 & 9, demonstrates competitive 3D awareness. For a qualitative analysis, we visualize the attention weights of UP-LVSM between the marked patch (red) in the reference view and each patch in the source view, as illustrated in Figure 8. The visualized weights demonstrate noticeable correspondence awareness, even compared to DINOv2 feature similarity. Both analyses confirm that UP-LVSM successfully develops a considerable degree of 3D awareness. See Appendix J.1 for detailed explanation.

**Probing Pose Accuracy** We further evaluate the accuracy of the latent poses produced by our Latent Plücker Learner by training a simple 2-layer MLP to map the latent poses to SE(3) space, supervised with pose annotations from RealEstate10K. We report the accuracy of mapped poses in Table 10, where our method achieves pose accuracy comparable to the concurrent Rayzer (Jiang et al., 2025). Qualitatively, we use t-SNE (Van der Maaten & Hinton, 2008) to visualize difference between the latent space and the real-world SE(3) space in Figure 9, demonstrating that the two spaces can align through a simple twisted domain transformation. See Appendix J.2 for details. Both results indicate that our model effectively learns the underlying 3D pose geometry using only 2D supervision.

| Model | Trans.@0.1 ↑ | Trans.@0.2 ↑ | Trans.@0.3 ↑ |
|---|---|---|---|
| Rayzer (Jiang et al., 2025) | 61.2 | 84.2 | 92.8 |
| UP-LVSM (Ours) | **71.3** | **89.2** | **96.4** |

| Model | Rot.@10° ↑ | Rot.@20° ↑ | Rot.@30° ↑ |
|---|---|---|---|
| Rayzer (Jiang et al., 2025) | **99.6** | **99.9** | **100** |
| UP-LVSM (Ours) | 98.4 | 99.6 | 99.8 |

Table 10: Accuracy of Mapping Latent Pose to SE(3).

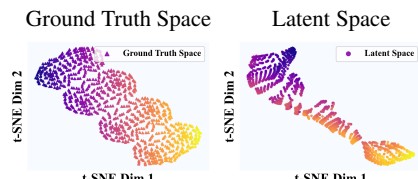

Figure 9: Visualization of the Learned Latent Space of Camera Poses.

**Camera Control** While UP-LVSM employs the Latent Plücker Learner to eliminate pose annotation dependence for improved scalability, the implicit nature of its estimated latent camera poses hinders the explicit control of the rendering view. However, it is easy to extend UP-LVSM for camera-controllable rendering in real-world scenarios by additionally learning a linear mapping from the SE(3) space to the learned latent space, as our *Latent Plücker Learner* effectively encourages the model to learn a meaningful manifold (Table 10 and Figure 9). After a regular training stage, we fine-tune UP-LVSM with this linear mapper using a small subset of posed data (1202 scenes in RealEstate10K, 1.8% of the training dataset). Table A demonstrates little

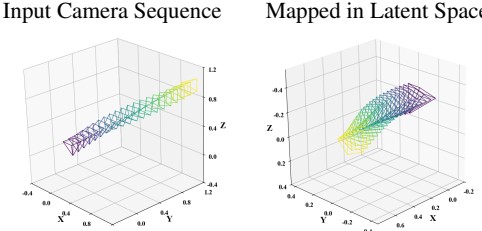

Figure 10: The linear transformation effectively maps input camera sequence into latent space, facilitating explicit camera control.

| Model | PSNR↑ | SSIM↑ | LPIPS↓ |
|---|---|---|---|
| UP-LVSM (w. Latent Plücker Learner) | 28.82 | 0.891 | 0.104 |
| UP-LVSM (w. Linear Mapper) | 28.41 | 0.886 | 0.110 |

Figure 11: NVS Performance with Mapped Poses.

performance degradation of this finetuning stage. We further visualize the linearly mapped poses in Figure B, providing evidence that our design effectively supports transformation between explicit SE(3) cameras and the latent ones, allowing human-specified camera sequences to be directly mapped to latent Plücker representations for controllability. More details are provided in Appendix D.

## 5 CONCLUSION

In this work, we revisit the field of feed-forward novel view synthesis through the lens of 3D knowledge dependency. We first highlight the need to reduce dependence on 3D knowledge by analyzing the scaling behaviors of state-of-the-art methods, revealing a key trend: methods with less 3D dependence accelerate dramatically as data scales—*the less you depend, the more you learn*. Building on this, we propose a novel NVS framework bypassing the need of explicit scene structure and camera pose annotations. By eliminating these 3D knowledge dependencies, our method leverages data scaling to foster implicit 3D awareness from 2D imagery, even outperforming the 3D knowledge-driven counterparts, thereby validating the effectiveness of our data-centric paradigm.

## REPRODUCIBILITY STATEMENT

The scalability analysis in Section 3, the training of our proposed method (UP-LVSM), and the experimental results in Section 4.3 & 4.4, are all reproducible. Details of the scalability analysis are provided in Section 3 and Appendix I.2 to ensure reproducibility. Details of training UP-LVSM are provided in Appendix I.1 with the architecture illustrated in Section 4.1. Details of investigation in Section 4.4 are provided in Appendix J. Furthermore, we will release code for the training and evaluation of our UP-LVSM to facilitate future research for the community.

## ACKNOWLEDGEMENT

This work is supported by the projects of Beijing Science and Technology Program (Z251100008125028).

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

# A    PROBLEM SETTING DETAILS

In this section, we provide additional details regarding the problem settings described in Section 3 of the main paper, where we present our analysis to support the hypothesis: *The less you depend, the more you learn*. Clarifying these settings is essential, as our analysis relies heavily on experimental results and comparative evaluations across state-of-the-art methods. While Figure 1 in the main paper illustrates the three problem settings we categorize, and Table 1 lists the corresponding methods, the definitions of these settings—such as their inputs, outputs, and evaluation protocols—are only briefly discussed for clarity. In this appendix, we elaborate on these aspects, with particular attention to distinctions between training, evaluation, and real-world deployment.

## A.1    POSED SETTING

The *posed* setting is the most straightforward scenario, assuming that pose information is always available. During both training and evaluation, the pose of the target view $\mathcal{P}_{\mathcal{T}}$ is provided by the dataset. In real-world applications, however, $\mathcal{P}_{\mathcal{T}}$ is determined by a user query, reflecting natural camera control behavior.

A critical aspect of this setting is that the view synthesis problem is inherently *pose-equivalent*: any given instance with poses $\mathcal{P}_{\mathcal{I}}, \mathcal{P}_{\mathcal{T}}$ is functionally equivalent to one with poses $\mathcal{H}'\mathcal{P}_{\mathcal{I}}, \mathcal{H}'\mathcal{P}_{\mathcal{T}}$, where $\mathcal{H}'$ is an arbitrary transformation in SE(3). To ensure pose-equivalence during training, a common practice is to apply **camera pose normalization**. This procedure treats the first input view as the reference, designating its pose as canonical and transforming all other camera poses from world coordinates into the canonical frame.

## A.2    POSED-TARGET SETTING

The *posed-target* setting introduces a subtle but important distinction. Unlike the posed setting, it does not require the input view poses $\mathcal{P}_{\mathcal{I}}$ for scene modeling, but it does require the target pose $\mathcal{P}_{\mathcal{T}}$ for view synthesis. Despite the pose-equivalence normalization discussed earlier, this setting inherently introduces ambiguity—specifically, how can a model reason about the spatial relationship between a posed target view and unposed input views?

To address this challenge, existing posed-target methods (Wang et al., 2021b; Fan et al., 2024; Ye et al., 2025) typically employ an **evaluation-time pose alignment** trick to ensure fair comparison on benchmarks such as RealEstate10K (Zhou et al., 2018). For example, in NoPoSplat (Ye et al., 2025), the model first estimates a 3D Gaussian Splatting (3DGS) representation in a canonical space from two unposed input views. This reconstructed 3DGS is then frozen, and the target camera pose is optimized at inference time so that the synthesized target view aligns as closely as possible with the ground truth image. It is important to note that this procedure is used solely for benchmark evaluation; in real-world applications, the target view pose is typically determined directly via user input, making such optimization unnecessary.

## A.3    UNPOSED SETTING

The *unposed* setting presents the most challenging scenario. Unlike the posed-target setting, where the target view pose $\mathcal{P}_{\mathcal{T}}$ is known and can guide view synthesis, the unposed setting assumes no pose information is available even during training, leaving explicit pose-based viewpoint control impossible.

To overcome this limitation, the early method, RUST (Sajjadi et al., 2023), employs a strategy similar to evaluation-time alignment, but adapted for training. Specifically, RUST introduces an implicit alignment mechanism by allowing the model to observe the ground truth target image $\tilde{\mathcal{T}}$ and learn to estimate its pose in a self-supervised manner, as depicted in Figure 1 of the main paper. Following this, our UP-LVSM framework introduces the *Latent Plücker Learner*, which estimates latent Plücker coordinates from the target view $\tilde{\mathcal{T}}$ and the scene latent $\mathcal{S}$. This design enables the model to infer the viewpoint from which to render, facilitating implicit alignment between the synthesized and ground truth target views for effective supervision. The concurrent work, Rayzer (Jiang et al., 2025), adopts a similar strategy by inferring the spatial relationship from multiple input and target views to predict each view's corresponding Plücker maps for view synthesis.

While such alignment techniques are effective during training and evaluation, they are unsuitable for real-world deployment, where the ground truth target view is unavailable. Unlike the posed and posed-target settings, where the target pose can be explicitly determined via human-specified camera sequences, the unposed setting relies on a learned, implicit pose space. This fundamentally limits direct, interpretable control and thereby weakens its practical applicability. To address this, our work prioritizes explicit camera control over implicit solutions (Sajjadi et al., 2023) or relative ones (Jiang et al., 2025), and proposes an effective strategy, which we detail in Appendix D.

## B    MORE RELATED WORK

**Feed-forward Novel View Synthesis**    Recent advancements in novel view synthesis using dense multi-view inputs have made significant progress (Mildenhall et al., 2020; Barron et al., 2022; Kerbl et al., 2023; Yu et al., 2024), but their reliance on explicit geometric cues limits applicability to unstructured observations. In contrast, generalizable methods aim to bypass computationally expensive per-scene optimization, typically by combining neural networks with 3D representations (Yu et al., 2021; Wang et al., 2021a; Du et al., 2023; Charatan et al., 2024; Chen et al., 2024; Xu et al., 2024; Zhang et al., 2024; Ye et al., 2025; Huang & Mikolajczyk, 2025). Another paradigm (Dosovitskiy et al., 2020; Rombach et al., 2021; Sajjadi et al., 2022; 2023; Suhail et al., 2022; Jin et al., 2025; Jiang et al., 2025) explores geometry-free solutions using feed-forward neural networks, with the recent methods (Jin et al., 2025; Jiang et al., 2025) achieving impressive results without explicit 3D bias.

**Multi-view Imagery Dataset**    Learning-based novel view synthesis approaches typically rely on large-scale datasets consisting of multi-view images and their corresponding camera parameters for training. Early datasets focused on object-level data (Chang et al., 2015; Reizenstein et al., 2021; Collins et al., 2022; Downs et al., 2022), while recent efforts (Yu et al., 2023; Deitke et al., 2023) have significantly expanded data scales. Meanwhile, several scene-level datasets (Dai et al., 2017; Chang et al., 2017; Li & Snavely, 2018; Yao et al., 2020; Li et al., 2021; Liu et al., 2021; Roberts et al., 2021; Yeshwanth et al., 2023; Ling et al., 2024; Tung et al., 2024) have been proposed to facilitate scene-level view synthesis. Among them, the RealEstate10K dataset (Zhou et al., 2018) has garnered significant attention due to its early release, open-source nature, and massive size, becoming a widely used training set and benchmark for recent generalizable view synthesis methods (Yu et al., 2021; Charatan et al., 2024; Chen et al., 2024; Ye et al., 2025; Jin et al., 2025).

**Pose-free View Synthesis**    Despite efforts to reduce dependence on input camera poses during inference (Fan et al., 2023; Smart et al., 2024; Ye et al., 2025; Zhang et al., 2025), generalizable novel view synthesis methods typically depend on posed data for training supervision, with few tackling the elimination of pose annotations. Early work (Sajjadi et al., 2023) pioneered the *really unposed* setting, bypassing pose dependence even during training. However, their solution struggles with high-fidelity rendering, and the latent pose representation remains difficult to interpret, making direct camera pose control challenging. In contrast, our data-centric framework harnesses scalability and the *Latent Plücker Learner* design, achieving rendering quality comparable to methods requiring pose input or supervision (Ye et al., 2025; Jin et al., 2025).

## C    MORE DISCUSSIONS ABOUT CONCURRENT WORK

As summarized in Table 1, there are two concurrent works also aiming for the unposed setting (Rayzer (Jiang et al., 2025) and SPFSplat (Huang & Mikolajczyk, 2025)).

**Rayzer**    While our design emphasizes camera control, the critical trend uncovered by our investigation is also confirmed by Rayzer (Jiang et al., 2025), a concurrent work with their focus on multiple sparse views ($N \geq 5$ instead of our $N = 2$), achieving promising results at the unposed setting, which bypasses 3D supervision. Most of our main paper experiments exclude this approach due to a difference in problem settings and the current lack of an official implementation. Despite this, in Table 11, we compare the results reported in its paper (Jiang et al., 2025) with our UP-LVSM performance on RealEstate10K. It is important to note that the Rayzer is trained with $N \geq 5$ input views, which differs from our $N = 2$.

**SPFSplat**    Another concurrent work, SPFSplat (Huang & Mikolajczyk, 2025), pushes the bias-driven approaches to the unposed setting by extending NoPoSplat (Ye et al., 2025) with a self-

supervised pose estimator. The training and evaluation settings of SPFSplat are consistent with ours, and we include its reported results in Table 11.

**Predictive Value of Our Findings**    Notably, the predictive value of our core findings is confirmed by the performance of these two methods. Rayzer, with its data-centric approach, significantly surpasses SPFSplat's bias-driven method in NVS metrics. We attribute this to SPFSplat's reliance on a predefined 3D representation (3DGS) and tailored, handcrafted rendering functions "creates a bottleneck at scale", as elaborated in Section 3. Furthermore, SPFSplat's performance (25.84 dB) aligns with the trend predicted by our scalability findings: as a bias-driven unposed method, it can outperform bias-driven posed-target methods (*e.g.*, NoPoSplat, 25.46 dB) but will be less competitive than the data-centric posed-target methods (*e.g.*, PT-LVSM, 26.00 dB).

| Method | Number of Input Views | Large Overlap | | | Medium Overlap | | | Small Overlap | | | Full Eval. | | |
| | | PSNR↑ | SSIM↑ | LPIPS↓ | PSNR↑ | SSIM↑ | LPIPS↓ | PSNR↑ | SSIM↑ | LPIPS↓ | PSNR↑ | SSIM↑ | LPIPS↓ |
|---|---|---|---|---|---|---|---|---|---|---|---|---|---|
| SPFSplat (Huang & Mikolajczyk, 2025) | 2 | 28.38 | 0.899 | 0.111 | 25.70 | 0.853 | 0.151 | 23.18 | 0.796 | 0.200 | 25.84 | 0.852 | 0.151 |
| Rayzer (Jiang et al., 2025) | 5 | N/A | N/A | N/A | N/A | N/A | N/A | N/A | N/A | N/A | 27.48 | 0.861 | 0.146 |
| UP-LVSM (Ours) | 2 | **29.51** | **0.901** | **0.098** | **26.93** | **0.852** | **0.132** | **24.54** | **0.796** | **0.174** | **28.82** | **0.891** | **0.104** |

Table 11: Quantitative Comparisons on RealEstate10K (Zhou et al., 2018). Following Ye et al. (2025), we conduct evaluations across different overlap levels. Our UP-LVSM consistently outperforms the concurrent approaches.

## D    CAMERA CONTROL

**Motivation**    While UP-LVSM employs the Latent Plücker Learner to eliminate pose annotation dependence for improved scalability, the implicit nature of its estimated latent camera poses hinders the explicit control of the rendering view. This might restrict its broad applicability. In this section, we demonstrate that it is easy to extend UP-LVSM for camera-controllable rendering in real-world scenarios by additionally learning a mapping from the `SE(3)` space to the learned latent space. Regarding camera intrinsics, we follow NoPoSplat (Ye et al., 2025) in assuming a known set of intrinsics for simplicity, while it is also feasible to extend the Latent Plücker Learner to accommodate learnable intrinsic.

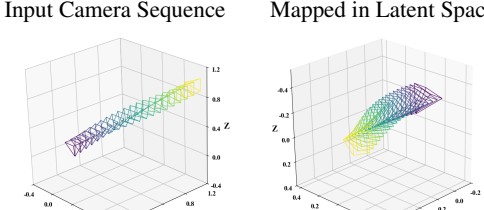

Input Camera Sequence          Mapped in Latent Space

Figure 12: The linear transformation effectively maps input camera sequence into latent space, facilitating explicit camera control.

| Model | PSNR↑ | SSIM↑ | LPIPS↓ |
|---|---|---|---|
| UP-LVSM (w. Latent Plücker Learner) | 28.82 | 0.891 | 0.104 |
| UP-LVSM (w. Linear Mapper) | 28.41 | 0.886 | 0.110 |

Table 12: NVS Performance with Mapped Poses.

**Implementation**    Specifically, we introduce a linear pose mapper parameterized by $(\mathbf{A} \in \mathbb{R}^{7 \times 7}, \mathbf{b} \in \mathbb{R}^7)$, which maps a real-world camera pose vector $\hat{\mathbf{C}} = \text{concat}(\hat{\mathbf{x}}, \hat{\mathbf{q}}) \in \mathbb{R}^7$ into its corresponding latent representation $\mathbf{C} = \text{concat}(\mathbf{x}, \mathbf{q}) = \mathbf{A}\hat{\mathbf{C}} + \mathbf{b} \in \mathbb{R}^7$. This latent camera pose is then used to generate the associated Plücker representation $\hat{\mathcal{P}}$. We fine-tune UP-LVSM with this linear mapper using a small subset of posed data (1202 scenes in RealEstate10K (Zhou et al., 2018), 1.8% of the training dataset). As described in Appendix A.1, we also apply camera pose normalization to reduce pose ambiguity. After fine-tuning, human-specified camera sequences can be directly mapped to latent Plücker representations for view synthesis. This fine-tuning introduces negligible impact on rendering quality, as evidenced in Figure 13 and Table 12.

**Discussion**    Earlier work (Sajjadi et al., 2023) also attempts to learn a latent pose, but typically fails to provide explicit camera control for view synthesis, primarily due to the uninterpretable nature of the learned pose latent. In contrast, our *Latent Plücker Learner* effectively encourages the model to learn a meaningful manifold as latent space, enabling controllability. This is evidenced by our investigation in Figure 9 of the main paper, where we visualize the learned latent Plücker space to validate that the model captures meaningful 3D pose space using only 2D supervision. We also visualize the linearly mapped poses in 12, providing evidence that our design effectively supports transformation between explicit `SE(3)` cameras and the latent ones.

The concurrent work, Rayzer (Jiang et al., 2025), also addresses camera control, but in a relative rather than explicit manner: it first estimates the camera poses of input views and then allows user-

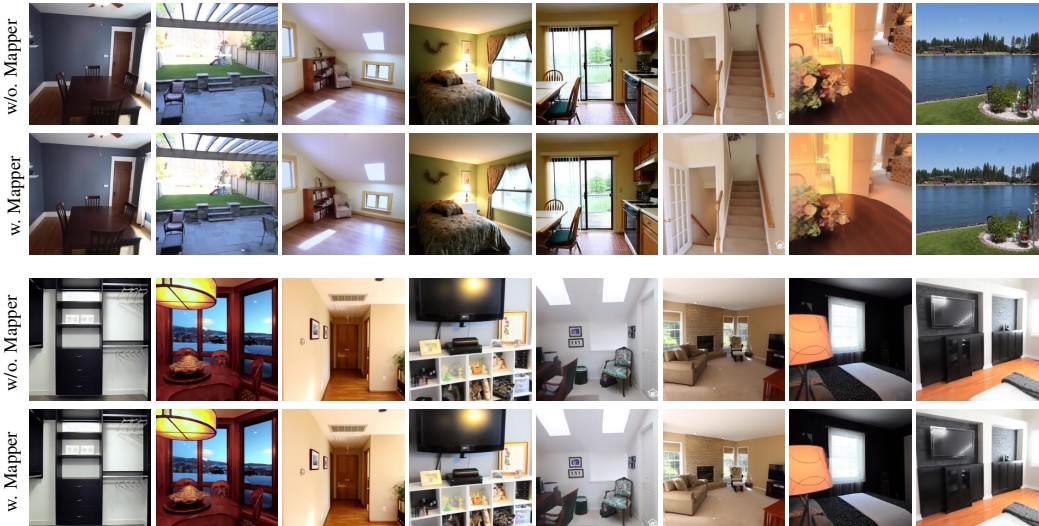

Figure 13: **Ablation Study of Camera Mapper.** The camera mapper fine-tuning introduces negligible impact on rendering quality.

specified interpolation between these poses. While effective, this strategy offers less flexibility than our approach. Additionally, it relies on multiple input views ($N \geq 5$) to infer spatial relationship, while our method only requires two views ($N = 2$).

## E    MORE DISCUSSION ABOUT SCALABILITY

In this section, we further explore the scalability analysis presented in Section 3 of the main paper. By conducting experiments on the synthetic dataset, we address a key question: *Why does the availability of camera poses limit scalability?*

**Background**    As discussed in Section 3.2, theoretically, camera poses provide additional information that acts as strong 3D cues. Thus, methods with access to camera poses (*e.g.*, LVSM (Jin et al., 2025)) should have a higher performance ceiling—one that significantly outperforms unposed methods like UP-LVSM. However, the curves in Figure 5 reveal an unexpected trend: our unposed method, UP-LVSM, achieves superior performance as the dataset scales. We attribute this unexpected trend to noise in pose annotations. Pose annotations in real-world datasets (Zhou et al., 2018; Yao et al., 2020; Yeshwanth et al., 2023; Ling et al., 2024) are typically generated by Structure-from-Motion (SfM) tools (Wu et al., 2011; Schonberger & Frahm, 2016)—which rely on geometric inductive biases—and these tools often introduce noise and inconsistencies. Relying on such noisy poses during training constitutes an indirect form of 3D knowledge dependence, creating a critical scalability bottleneck.

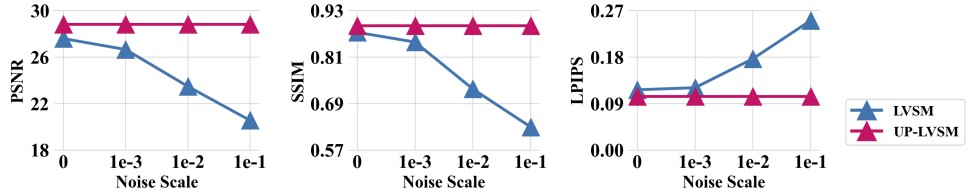

Figure 14: **Performance of Methods Trained with Noisy Poses.** Different levels of Gaussian noise ($\sigma^2 = 0.001, 0.01, 0.1$) were added to the rotation (in quaternion form) and translation components of the poses in training data. While UP-LVSM remains agnostic to noisy poses, LVSM experiences significant degradation with increasing noise levels, exhibiting sensitivity even to small amounts of noise (0.001).

**Investigation**    To validate this, we investigate the impact of pose noise on LVSM performance, with experimental results presented in Figure 14. These results show that even a small amount of noise significantly affects LVSM's performance. This provides partial insight into the trend observed in Figure 4 of the main paper, where unposed methods (*e.g.*, UP-LVSM) eventually outperform

those that depend on poses (*e.g.*, LVSM), even when the latter have access to additional information. Moreover, we additionally train LVSM and UP-LVSM on the Objaverse dataset (Deitke et al., 2023), where the synthetic data will comprise no pose noise. As demonstrated in Table 13, given ground truth camera poses, pose-dependent methods like LVSM begin to achieve superior performance as expected, providing strong evidence to support our claims. Moreover, under ground truth poses, LVSM and UP-LVSM exhibit similar scalability, indicating that it is unreliable pose annotations from SfM that limit scalability and lead to a lower performance ceiling.

| Method | $\mathcal{P}_{\mathcal{I}}$-free | NVS performance (PSNR↑ / SSIM↑ / LPIPS↓) | | | | ΔPSNR / ΔSSIM / ΔLPIPS |
|---|---|---|---|---|---|---|
| | | 2K | 8K | 32K | 128K | Avg. Gain ↑ |
| LVSM | ✗ | 24.58 / 0.814 / 0.177 | 28.90 / 0.887 / 0.106 | 29.63 / 0.898 / 0.096 | 30.22 / 0.906 / 0.087 | 1.77 / 0.029 / 0.028 |
| UP-LVSM | ✓ | 19.96 / 0.712 / 0.403 | 23.38 / 0.773 / 0.275 | 26.02 / 0.827 / 0.158 | 26.12 / 0.829 / 0.156 | 2.11 / 0.040 / 0.086 |

Table 13: Qualitative Results on Objaverse (Deitke et al., 2023).

## F   MORE RESULTS

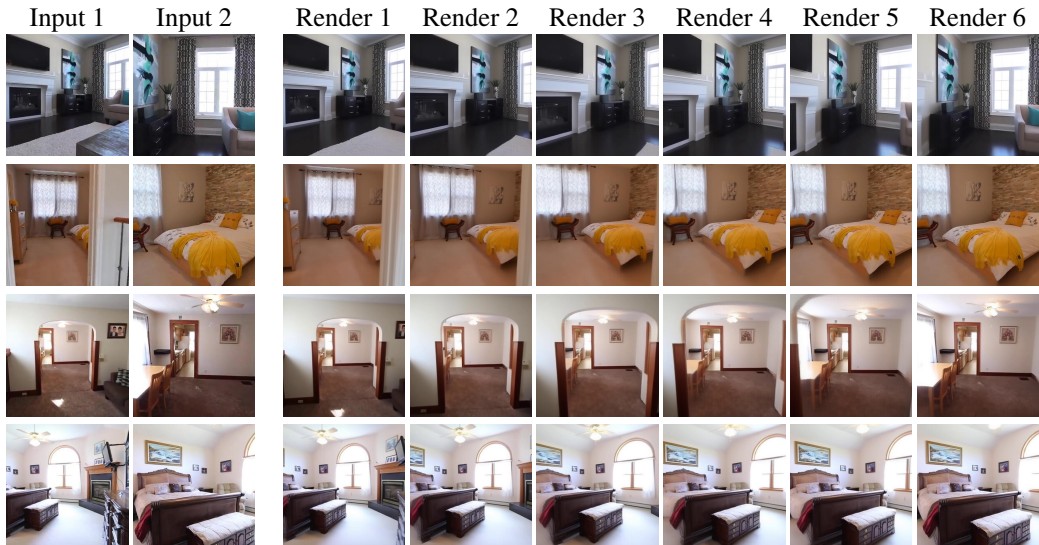

Figure 15: More Indoor Results of UP-LVSM.

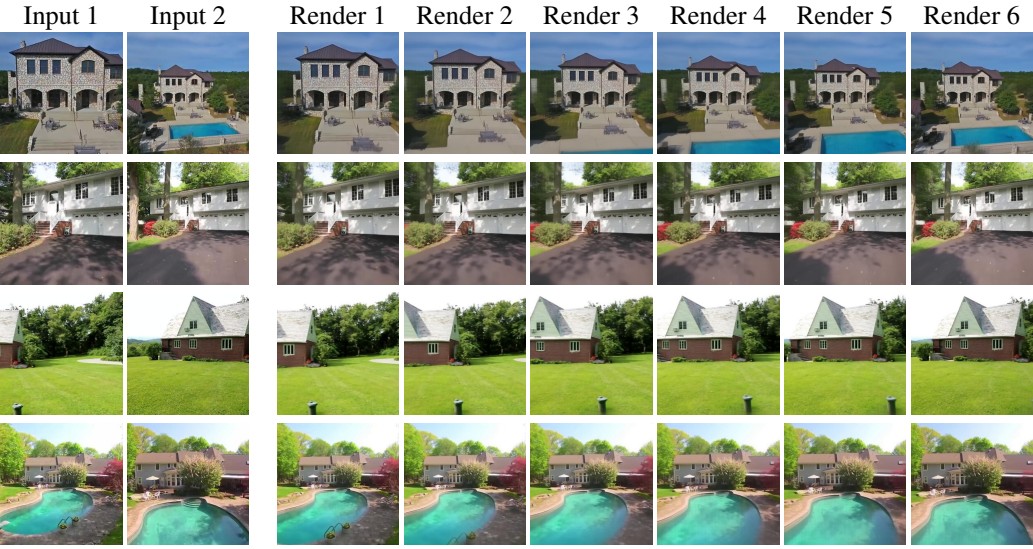

Figure 16: More Outdoor Results of UP-LVSM.

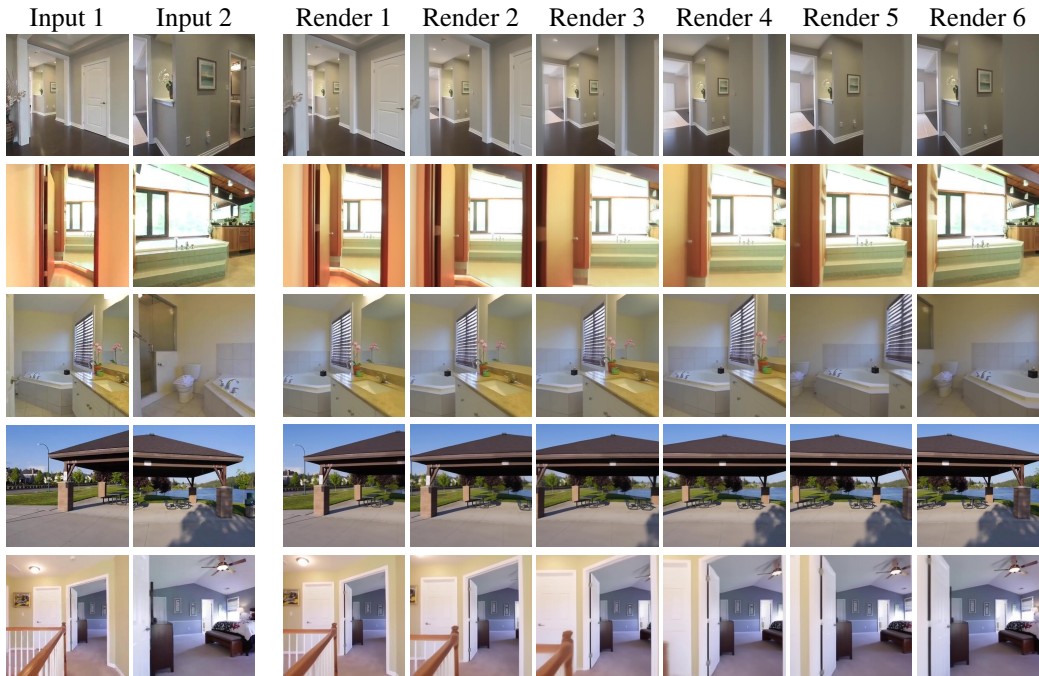

Figure 17: More Challenging Results of UP-LVSM (Little Overlapped Inputs).

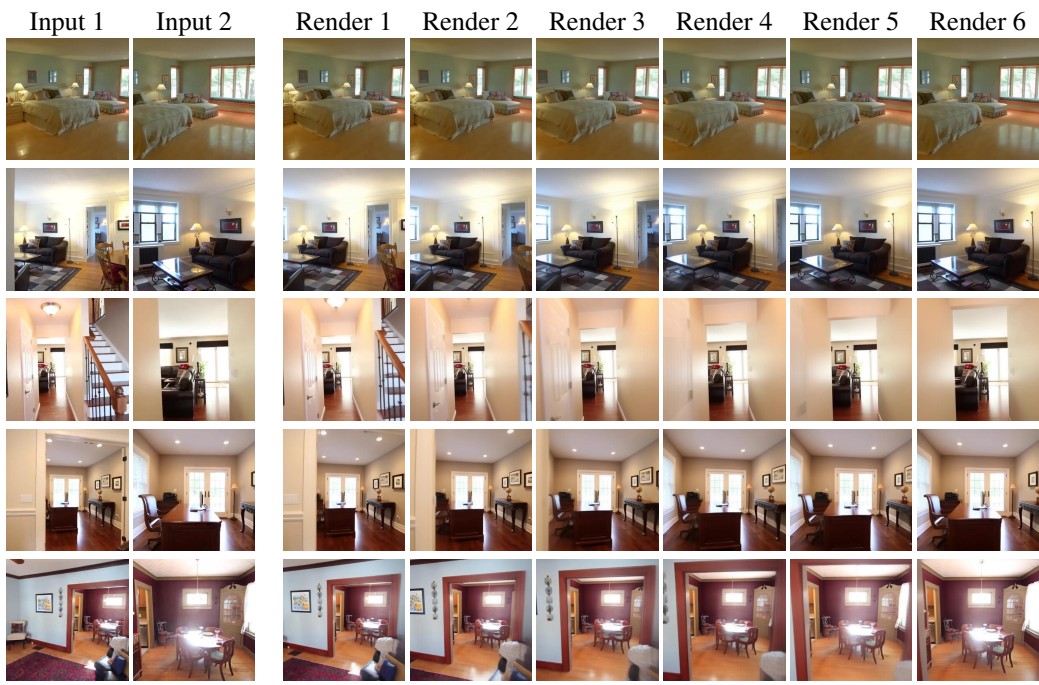

Figure 18: More Challenging Results of UP-LVSM (Complicated Lighting Effects).

## G  MORE DISCUSSION ABOUT EXTRAPOLATION

All the view synthesis results in the main paper or the supplementary video are mainly interpolation between inputs views. This is because currently, most existing generalizable novel view synthesis (NVS) methods are good at interpolation-style NVS, but perform much worse for extrapolation, as extrapolation is actually guessing what the whole scenes look like from partial observation, thereby indeed requiring generative modeling techniques (Wewer et al., 2024). Our methods suffer from the similar problem. Following extrapolation evaluation principles in previous work (Wewer

et al., 2024), we measure the extrapolation performance of existing methods. Table 14 compares the interpolation and extrapolation performance, demonstrating fundamental limitations of these generalizable methods.

| Evaluation | Metric | PixelNeRF | PixelSplat | MVSplat | LVSM | NoPoSplat | Ours (PT-LVSM) | Ours (UP-LVSM) |
|---|---|---|---|---|---|---|---|---|
| Interpolation | PSNR ↑ | 20.33 | 25.51 | 26.45 | 27.60 | 25.46 | 26.00 | 28.82 |
| | SSIM ↑ | 0.572 | 0.867 | 0.874 | 0.874 | 0.854 | 0.825 | 0.891 |
| | LPIPS ↓ | 0.549 | 0.126 | 0.123 | 0.117 | 0.137 | 0.135 | 0.104 |
| Extrapolation | PSNR ↑ | 19.96 | 21.19 | 20.00 | 23.80 | 22.42 | 19.18 | 23.82 |
| | SSIM ↑ | 0.572 | 0.799 | 0.787 | 0.795 | 0.786 | 0.611 | 0.760 |
| | LPIPS ↓ | 0.568 | 0.196 | 0.205 | 0.168 | 0.201 | 0.282 | 0.185 |

Table 14: **Quantitative Comparisons.** Existing methods perform much worse for extrapolation.

## H    MORE DISCUSSION ABOUT LIMITATION

**Scalability Analysis**    In the main paper (Section 3.3), we validate the key trend that reducing dependence on 3D knowledge enhances scalability on three real-world datasets, *i.e.*, RealEstate10K (Zhou et al., 2018), DL3DV (Ling et al., 2024), and ACID (Liu et al., 2021). Of these, the largest is RealEstate10K, which comprises 66K scenes. An open question remains: if data scale continues to grow beyond 66K scenes, can UP-LVSM sustain this performance trend? In other words, at what point will the performance curves in Figure 5 begin to saturate? Although we have observed this when validating the same trend on the synthetic Objaverse dataset (Deitke et al., 2023) (see Appendix E), we cannot conduct this experiment on larger real-world datasets primarily because open-source NVS data remains limited. But we believe our UP-LVSM, benefiting from its pose-free training, can leverage large-scale video datasets to effectively extend training quantities, not limited to NVS datasets, with a primary challenge lying in that these datasets may lack strong camera motion and contain dynamic scenes, which would necessitate extensive data cleaning.

**Methodology**    From a methodological standpoint, our UP-LVSM is still based on DINOv2 (Oquab et al., 2023) and inherits its limitation of a relatively large patch size of 14 (compared to LVSM's patch size of 8). The large patch size hinders fine-granularity image synthesis, leading to blurring artifacts in richly textured areas. As demonstrated in LVSM (Jin et al., 2025), smaller patch sizes lead to more competitive performance at the cost of increased training time and higher CUDA memory usage. Striking a balance between performance and training cost, particularly through improvements to the network architecture, is an important avenue for future exploration. Furthermore, we observe that increasing the amount of training data increases the risk of gradient explosion. While we mitigate this issue by adopting QKNorm (Henry et al., 2020) as in LVSM, addressing this issue more effectively, particularly when scaling to larger datasets, will be crucial in future work.

## I    IMPLEMENTATION DETAILS

In this section, we provide implementation details for the methods compared in the main paper, including network architectures and training hyperparameters.

### I.1    PT-LVSM & UP-LVSM

Following prior works (Sajjadi et al., 2022; Zhang et al., 2024; Jin et al., 2025), the core architecture of PT-LVSM is composed entirely of Transformer layers (Vaswani et al., 2017). Unlike previous implementations that train the encoder from scratch, we incorporate a pretrained DINOv2 encoder (Oquab et al., 2023) to enhance training stability, particularly in the early stages, due to the absence of input pose annotations. The Transformer component adopts a decoder-only architecture, as in LVSM (Jin et al., 2025), comprising 24 layers. Each multi-head attention layer includes 12 heads, each with 64-dimensional feature embeddings. The entire model, including both the Transformers and the DINOv2 encoder, is jointly optimized with a learning rate of 0.0004.

Training is conducted on the full RealEstate10K dataset (Zhou et al., 2018) using 8 NVIDIA A100 GPUs, with a batch size of 16 per GPU. Training for 100K steps takes approximately 60 hours. The loss function used is $\mathcal{L} = \mathrm{MSE}(\mathcal{T}, \tilde{\mathcal{T}}) + \lambda \mathrm{Perceptual}(\mathcal{T}, \tilde{\mathcal{T}})$, where $\lambda = 0.5$, and $\mathrm{Perceptual}$

denotes the perceptual loss introduced in (Johnson et al., 2016). For numerical stability, we follow LVSM (Jin et al., 2025) in employing QKNorm (Henry et al., 2020) to mitigate the risk of gradient explosion.

The architecture of UP-LVSM differs slightly due to its encoder-decoder structure and latent Plücker representation. The encoder comprises a DINOv2 backbone followed by 6 Transformer layers. The decoder consists of 14 Transformer layers. Additionally, the Latent Plücker Learner uses a DINOv2 encoder followed by a 4-layer Transformer. Note that our number of layers is set equal to LVSM and PT-LVSM for fair comparisons under the same level of parameter amount. All other training settings are consistent with those of PT-LVSM. After pretraining, UP-LVSM is fine-tuned to support camera-controllable rendering using a linear mapper. This fine-tuning is performed on the *little* subset of the RealEstate10K dataset (1202 scenes, 1.8% of the full set) with a learning rate of 0.0001, requiring approximately 4 hours for 8K steps. During this stage, the ground truth target image is no longer provided; instead, the latent Plücker is generated via the linear mapper from the ground truth target pose, rather than from the Latent Plücker Learner.

For both PT-LVSM and UP-LVSM, we adopt camera pose normalization mentioned in Appendix A.1 to designate the pose of the first input view as canonical. However, this conflicts with the permutation-invariant nature of the Transformer, where the first input view should be recognized as special, but the Transformer inherently treats all inputs equally. To this end, we assign special significance to the first view by adding a linearly projected canonical Plücker onto its DINOv2 image tokens. The ablation study demonstrates our model cannot converge when trained without this trick, validating its effectiveness.

Lastly, due to the DINOv2 encoder's requirement that input dimensions be divisible by the patch size of 14, we rescale RealEstate10K images to a resolution of $224 \times 224$, rather than the more commonly used $256 \times 256$. Following the approach in LVSM, we first train at low resolution (e.g., $224 \times 224$), and then fine-tune on higher resolutions such as $518 \times 518$ to better adapt the model to high-resolution rendering. However, for the experiments reported in the main paper, we standardize all evaluations to the $224 \times 224$ setting, including all baseline comparisons.

## I.2 BASELINES

For all baseline methods evaluated in the main paper—PixelNeRF (Yu et al., 2021), Pixel-Splat (Charatan et al., 2024), MVSplat (Chen et al., 2024), LVSM (Jin et al., 2025), and NoPoSplat (Ye et al., 2025)—we use the original training configurations provided in their respective official repositories. Since PixelNeRF does not provide official configurations for the RealEstate10K dataset, we adapt its official code to work with this dataset and successfully reproduce the performance reported in (Ye et al., 2025; Charatan et al., 2024). All other methods include official support for the RealEstate10K dataset, requiring no modification aside from rescaling the input images to $224 \times 224$ (consistent with our setup as described above).

## I.3 OBJECT-LEVEL TRAINING

| Method | | Perceptual Weight $\lambda$ | Background | Rendering Views | Joint Training | Performance (PSNR↑) |
|---|---|---|---|---|---|---|
| | (a) | 0.5 | White | 24, Sparse | No | Not Converged. |
| | (b) | 0.2 | White | 24, Sparse | No | Not Converged. |
| | (c) | 0.5 | Gray | 24, Sparse | No | Not Converged. |
| UP-LVSM | (d) | 0.2 | Gray | 24, Sparse | No | 23.38 |
| | (e) | 0.2 | Gray | 128, Sequantial | No | 24.95 |
| | (f) | 0.2 | Gray | 128, Sequantial | Yes | 24.37 |
| | (g) | 0.2 | Gray | 24, Sparse | Yes | Not Converged. |

Table 15: We use a small amount (8K) of object-level data to investigate different training strategy, including varying background color for alpha compositing, different strategies of view rendering, and whether trained jointly with 8K scene-level data from RealEstate10K.

For object-level experiments (Tables 4 and 13) on the Objaverse dataset (Deitke et al., 2023), we follow prior works (Jin et al., 2025) to train our method and the baselines under a different setting, as demonstrated in Table B (d). We first prepared 136K objects from Objaverse, with each rendered by Blender from 24 randomly sampled views. During training, we sample 4 input views and 8 target views at each step, in contrast to the scene-level training's 2 input views and 6 target views. To avoid

instability, we use a perceptual loss weight $\lambda$ of 0.2 instead of 0.5. The experimental results are shown in Table B, where we discuss the influence of different strategies on training stability.

## J  INVESTIGATION DETAILS

This section provides additional details for the analysis presented in Section 4.4 of the main paper, including attention weight analysis and latent Plücker space analysis.

### J.1  ATTENTION WEIGHT ANALYSIS

We elaborate on Figure 8 of the main paper by visualizing patch-wise attention weights to illustrate that our model performs spatial reasoning and captures cross-view correspondences. Below, we describe the process in detail.

Consider an input image resolution of 224. Following the DINOv2 architecture, which uses a patch size of 14, each input image $\mathcal{I} \in \mathbb{R}^{B \times N \times 224 \times 224 \times 3}$ is converted into feature tokens $\mathbf{D} \in \mathbb{R}^{B \times N \times 16 \times 16 \times 768}$, where $B$ is the batch size and $N$ is the number of input views. These tokens are then flattened to $\mathbf{D}' \in \mathbb{R}^{B \times 256N \times 768}$ and passed through the Transformer layers in the encoder.

We examine the attention weights $\mathbf{W} \in \mathbb{R}^{B \times 256N \times 256N}$ from the final Transformer layer, where each element represents the attention between pairs of input patches across all views. In the case where $N = 2$ and $B = 1$, the bottom-left $256 \times 256$ block of $\mathbf{W}$, denoted as $\mathbf{W}' \in \mathbb{R}^{256 \times 256}$, corresponds to the cross-view attention between the two input views. Specifically, each element $\mathbf{W}(i, j)$ indicates the attention weight from the $i$-th patch of the first view to the $j$-th patch of the second view.

While we use the *viridis* colormap to visualize the attention weights $\mathbf{W}$ in Figure 8 of the main paper, we also visualize the DINOv2 token similarity to verify that the model learns cross-view correspondence during training, rather than relying solely on the pretrained DINOv2 encoder's inherent capabilities. Specifically, given DINOv2 tokens $\mathbf{D}'_1, \mathbf{D}'_2 \in \mathbb{R}^{256 \times 768}$ from two views, we compute their cosine similarity along the feature dimension to obtain $\mathbf{S} \in \mathbb{R}^{256 \times 256}$. For clearer visualization, $\mathbf{S}$ is normalized as $\mathbf{S}' = \frac{\mathbf{S} - \min(\mathbf{S})}{\max(\mathbf{S}) - \min(\mathbf{S})}$.

### J.2  LATENT PLÜCKER ANALYSIS

This section provides a detailed explanation of the analysis presented in Figure 9 of the main paper. Camera poses are represented as 7-dimensional vectors, with the first 3 dimensions encoding translation and the last 4 representing rotation as a quaternion. Given $N$ pairs of camera poses $\mathbf{C}_{\text{latent}} \in \mathbb{R}^{N \times 7}$ and $\mathbf{C}_{\text{real}} \in \mathbb{R}^{N \times 7}$, we concatenate them to form $\mathbf{C}_{\text{all}} \in \mathbb{R}^{2N \times 7}$. We then apply t-SNE (Van der Maaten & Hinton, 2008) to project this 7-dimensional space into two dimensions, yielding $\mathbf{C}'_{\text{latent}}, \mathbf{C}'_{\text{real}} \in \mathbb{R}^{N \times 2}$ after splitting. In Figure 9, these 2D points are visualized accordingly, with corresponding pairs colored identically.

## K  THE USE OF LARGE LANGUAGE MODELS (LLMs)

We used Large Language Models (LLMs) as writing assistants for this work. They are used only for text polishing, grammar checking, and sentence-level rephrasing. The core method development in this work does not involve LLMs as any important, original, or non-standard components.

