# OpenReview forum: "The Less You Depend, The More You Learn: Synthesizing Novel Views from Sparse, Unposed Images without Any 3D Knowledge"
_ICLR.cc/2026/Conference — ICLR 2026 Poster_

### Official Review · Reviewer_KQEM · 2025-10-28

**Soundness:** 3
**Presentation:** 3
**Contribution:** 3
**Rating:** 6
**Confidence:** 4

**Summary:**

The authors study generalizable novel view synthesis (NVS) from sparse, unposed 2D images and show that methods with less explicit 3D knowledge improve faster with larger data; they propose a framework that eliminates 3D inductive bias and pose annotations and learns implicit 3D awareness from 2D images, reporting photorealistic and 3D-consistent novel views comparable to methods that use posed inputs

**Strengths:**

1. Clear empirical trend discovery, identifies and quantifies the data-scaling trend: models that depend less on explicit 3D knowledge improve faster as data scales, motivating a data-centric alternative to heavy 3D priors.

2. The framework removes both explicit 3D architectural bias and pose supervision, which, if robust, simplifies pipelines and broadens applicability to unposed datasets.

3. Practical impact in terms of enabling NVS without pose labels lowers annotation cost and allows training on large, in-the-wild photo collections.

4. Empirical results claim parity with posed-input methods, suggesting the approach is not only elegant but effective on benchmarks.

5. The paper reframes the trade-off between inductive bias and data scale, providing a clear hypothesis for future work and dataset collection strategies

**Weaknesses:**

1. The method’s advantages hinge on data scaling; performance in low-data regimes or niche domains may degrade relative to 3D-aware methods.

2. Without explicit 3D representations or poses, ensuring geometric consistency across large camera motions or severe occlusions can be fragile; failure modes and long-range consistency are likely under-explored.

3. Reported parity with posed methods may depend on specific datasets or evaluation metrics; results on highly diverse, real-world scenes or metric 3D accuracy may be weaker.

4. Removing explicit 3D structure reduces model interpretability and makes diagnosing geometric errors or dataset biases harder.

5. Achieving strong performance via data scaling may require substantial compute and careful curricula; the practical resource requirements may be high.

6. Learning implicit 3D from 2D alone risks encoding photographic regularities that do not generalize to different capture conditions or object categories.

**Questions:**

1. At what dataset size and diversity does the “less 3D knowledge” model overtake 3D-driven counterparts? Is there a measurable crossover point per dataset type?

2. Which camera motions, occlusion patterns, or scene topologies cause the model to produce inconsistent geometry or view synthesis artifacts?

3. How does the method perform on metric 3D/geometry benchmarks (e.g., depth/pose consistency) compared with explicit 3D methods?

4. Does training on large web-scale data produce robust synthesis on specialized domains or is fine-tuning with some 3D bias still necessary?

5. Can hybrid approaches (weak 3D priors, self-supervised pose signals) such as HawkI reduce data needs while retaining benefits of the minimalist design?

6. What are the training compute, memory, and inference-time costs relative to NeRF-style or other 3D-aware baselines?

7. Which minimal inductive elements (if any) materially help (positional encoding, multi-view consistency losses, rendering modules) and which are redundant?

8. Does reliance on large web data introduce unwanted biases in scene content, lighting, or demographic representation that affect downstream uses?

9. If we train on large in-the-wild data and evaluate on narrow-domain datasets to measure generalization and need for adaptation, how does it work?

10. It will be good to add experiments with light-weight 3D cues (approximate poses, sparse depth) to quantify trade-offs between inductive bias and data size.

11. It will be good to publish FLOPs, GPU hours, and inference latency to clarify practical deployment feasibility

---

> ### Author Response · Authors · 2025-11-21
> **Rebuttal for Reviewer KQEM (1/3)**
>
> We thank Reviewer KQEM for the thoughtful and wide-ranging questions. Below, we provide a detailed explanation for each:
>
> ---
>
>
> ### **[W1] Concerns about performance in low-data regimes**
>
> Thanks for raising this point. As analyzed in Section 3, UP-LVSM is not designed for low-data regimes and does not perform optimally there. However, we believe our method remains valuable because it achieves state-of-the-art performance at the commonly used training scales in feed-forward NVS, where data availability is typically sufficient.
>
> ---
>
> ### **[W2] Robustness for Large Camera Motion**
>
> Thanks for raising this issue. The reviewer is concerned that, without explicit 3D, our model might be fragile under large motions, occlusions, or complex scene topologies. We would like to clarify that our method performs well under these challenging cases, primarily due to its pose-agnostic nature. Specifically,
>
> 1. Pose-supervised methods rely on camera annotations estimated by noisy SfM. In challenging regimes (large parallax, strong occlusions, complex lighting), they are often unreliable, and explicit 3D pipelines (e.g., NeRF/3DGS) inherit and amplify such errors.
> 2. Pose-free methods remain robust. As discussed in Appendix E, pose-supervised baselines degrade sharply as camera noise increases, whereas UP-LVSM remains unchanged, precisely because it does **not** depend on camera annotations.
> 3. Empirically, UP-LVSM delivers robust results under difficult camera motions or large occlusions. Visuals under such conditions are provided in Appendix F.
>
>
> Moreover, we now include additional long-sequence visualizations ( `rebuttal.mp4` ) in the supplementary material, demonstrating that our method maintains robust consistency over long trajectories.
>
> We hope this addresses the reviewer’s concern.
>
> ---
>
> ### **[W3] Real-world diverse scenes**
>
> Regarding the concern that UP-LVSM may rely on specific datasets, we would like to clarify:
>
> 1. **Dataset diversity.** We train and evaluate on Objaverse, RealEstate10K, DL3DV, and ACID, which together cover large-scale real-world scenarios with great diversity. As shown in Sec. 4.3, UP-LVSM consistently outperforms prior methods across these datasets, rather than only on a single benchmark.
> 2. **Generalizability.** UP-LVSM exhibits strong zero-shot generalization. As shown in Table 6, zero-shot evaluation of RE10K→ACID even surpassed ACID→ACID, indicating that our gains are not tied to a narrow data distribution but benefit from scaling with great generalizability.
>
> We hope this could justify the robustness of the proposed framework.
>
> ---
>
> ### **[W4] Reduced interpretability without explicit 3D structure.**
>
> As the reviewer insightfully points out, our implicit scene structure could lack interpretability compared to its explicit counterparts. We attribute this to a fundamental paradigm-level trade-off, and there’s no free lunch:
>
> 1. **Explicit scene structure models** (e.g., 3DGS, meshes) provide direct, interpretable geometry, but are tightly coupled to SfM quality and hand-crafted rendering pipelines, and can degrade significantly when camera estimates are noisy.
> 2. **Implicit, data-centric models** like UP-LVSM sacrifice explicit 3D structure in exchange for robustness to pose noise, a self-supervised pipeline, and better scalability with data.
>
> To partially bridge the interpretability gap, we already probe the learned 3D awareness in several ways, as shown in Sec. 4.3/4.4 and Appendix D. These results demonstrate that, although UP-LVSM does not output explicit 3D geometry, it still learns a meaningful and quantitatively verifiable notion of 3D structure.
>
> ---
>
> ### **[W5] Resource requirements for training**
>
> The reviewer is concerned about the potential training cost. As described in Appendix I.1, we rely on 8 A100 GPUs for training, at the same computational resource level as prior works like NoPoSplat and LVSM. We hope this could justify that our method does not rely on significantly richer computational resources.
>
> ---
>
> ### **[W6] Robustness to photographic regularities**
>
> We would like to clarify that by training and evaluating on diverse datasets, such as Objaverse, RealEstate10K, DL3DV, and ACID, the data-centric framework naturally allows UP-LVSM to handle complex high-order appearance effects (e.g., reflections, non-Lambertian materials).

---

> ### Author Response · Authors · 2025-11-21
> **Rebuttal for Reviewer KQEM (2/3)**
>
> ### **[Q1] Cross-over point when the “less 3D knowledge” model overtake 3D-driven counterparts**
>
> As the reviewer points out, a crossover point exists, after which data-centric models continue to improve while 3D-biased models saturate. Empirically, the cross-over typically occurs between the 16K and 64K-scene regime on RealEstate10K, and varies with different metrics and datasets.
>
> ---
>
> ### **[Q2] Failure modes**
>
> As a non-generative model for view synthesis, our main failure modes arise in extrapolation, as discussed in Appendix G. Similar observations could also be found in Appendix E from RayZer [1].
>
> > [1] RayZer: A Self-supervised Large View Synthesis Model
> >
>
> ---
>
> ### **[Q3] Metric 3D benchmark**
>
> Thanks for raising this issue. We agree that assessing 3D awareness can provide valuable insights. However, as UP-LVSM is an implicit, token-based model rather than a depth- or field-predicting one, direct comparison to explicit 3D methods on metric depth or geometry is not applicable. Instead, as part of the implicit–explicit paradigm trade-off, we rely on indirect evidence obtained by probing geometry and camera awareness, as discussed in Sec. 4.4 and Appendix D.
>
> ---
>
> ### **[Q4] Whether fine-tuning with some 3D bias is still necessary**
>
> To our best understanding, for training UP-LVSM, **data scale and diversity** are the dominant factors. Given large data quantity, fine-tuning with some 3D bias is not necessary. However, we appreciate the reviewer's insightful questions and agree that it is promising to further investigate the impact of fine-tuning with additional 3D bias. We believe this is potentially one of future research directions.
>
> ### **[Q5] Hybrid approaches**
>
> We appreciate the suggestion. Our current work intentionally explores the extreme minimalist end of the spectrum (no explicit 3D representation, no pose annotations) to isolate the effect of data scaling. That said, hybrid approaches are indeed promising directions to reduce data demands while retaining many of the benefits of our design. For example:
>
> 1. Using self-supervised homography/epipolar cues as weak 3D signals,
> 2. Leveraging synthetic data with controllable 3D priors, or
> 3. Distilling from 3D-aware generative or reconstruction models.
>
> Such designs could potentially reduce data requirements while preserving many of the benefits of our data-centric formulation. We have not explored these hybrids in the current submission, but we agree they form a compelling direction for future work and will highlight this in the outlook section.
>
> ### **[Q6] Training Compute, Memory, and Inference-time Costs**
>
> Thanks for raising this issue. We report the typical training/inference costs as follows:
>
> - UP-LVSM
>   - Training: 8xA100 with 80GB CUDA memory per GPU for 60 hours;
>   - Inference: 1x4090 with 24GB CUDA memory.
>
> - PixelNeRF
>   - Training: 1x4090 with 24GB CUDA memory per GPU for 150 hours;
>   - Inference 1x4090 with 24GB CUDA memory.
>
> - NoPoSplat
>   - Training: 8xA100 with 80GB CUDA memory per GPU for 6 hours;
>   - Inference: 1x4090 with 24GB CUDA memory.
>
> We hope these additional details could be useful.

---

> ### Author Response · Authors · 2025-11-21
> **Rebuttal for Reviewer KQEM (3/3)**
>
> ### **[Q7] Minimal inductive elements**
>
> In our experiments, we observe the following:
>
> 1. **Helpful:** The Plucker-style embedding, if interpreted as a geometric positional encoding, is crucial for stable viewpoint conditioning and clearly improves performance, as shown in the ablation studies (Table 7).
> 2. **Not used:** We do **not** use explicit multi-view consistency losses; the supervision comes purely from the NVS objective.
> 3. **Redundant:** Explicit rendering modules, such as a Gaussian-splatting renderer, are not necessary for UP-LVSM.
>
> We hope this could be helpful and are open to answering further questions.
>
> ---
>
> ### **[Q8] Dataset biases**
>
> While we follow prior works to leverage huge web data like RealEstate10K, we agree with the reviewer on these insightful points, that neural networks trained on large in-the-wild data could inevitably be exposed to biases in scene content, lighting, and demographic representation. We believe this could be mitigated by data cleaning or specific techniques like fairness-aware algorithms. We thank the reviewer for pointing out this point.
>
> ---
>
> ### **[Q9] Evaluation on narrow-domain datasets**
>
> We appreciate the valuable question and would like to have a detailed explanation. Basically, when the domain shift is not too extreme (e.g., regular photos vs. similar real-world scenes, rather than natural images vs. medical X-rays), our proposed framework could demonstrate good zero-shot generalizability. As evidenced in Table 6, UP-LVSM generalizes well from RE10K to ACID. However, for more specialized or highly out-of-distribution domains, as the reviewer constructively points out, some form of adaptation or fine-tuning may still be beneficial.
>
> ---
>
> ### **[Q10] Lightweight 3D cues**
>
> We agree that lightweight 3D cues could be useful. In our experiments, the **source and quality** of these cues are crucial: when they come from noisy SfM, they tend to be harmful rather than helpful. This is reflected in Table 7, where training with SfM camera annotations performs worse than our self-supervised latent Plücker. This suggests that truly clean ground-truth poses or sparse depth could, in principle, help reduce data requirements, but such noise-free 3D labels are rarely available at real-world scale—the regime our method is designed for.
>
>
> ### **[Q11] Training/inference statistics**
>
> Thanks for raising this issue. We agree that reporting these numbers is important for assessing practical feasibility. Most training statistics for UP-LVSM are already provided in Appendix I. For the other mentioned metrics, we report as follows:
>
> - Training: 8xA100 with 100GB physical memory.
> - GPU hours (training): about 500 hours (about 60 hours per GPU).
> - Inference (6 target views): 275.67 GFLOPS with 56.2ms latency.
>
> We hope this could help enhance practical deployment feasibility. Further, we commit to releasing the code of UP-LVSM for reproducibility.
>
> ---
>
> ## **Summary**
>
> Again, we appreciate Reviewer KQEM for their effort during the review period. We hope our explanation addresses the concerns, and we are happy to clarify any further questions.

---

### Official Review · Reviewer_n5sM · 2025-10-31

**Soundness:** 3
**Presentation:** 4
**Contribution:** 4
**Rating:** 8
**Confidence:** 4

**Summary:**

This paper investigates the trade-off between 3D bias-driven and data-centric design philosophies for Novel View Synthesis (NVS). The central hypothesis is "the less you depend, the more you learn": methods with weaker 3D dependencies exhibit superior scalability, accelerating in performance as training data increases.

The paper tests this hypothesis by conducting a systematic scalability analysis comparing existing methods (MVSplat, LVSM, NoPoSplat) on training subsets of varying sizes. They find that data-centric methods show greater performance gains with more data.

Building on this insight, the paper proposes UP-LVSM, a novel, data-centric framework that operates in a fully unposed setting. This method learns to synthesize novel views from sparse images without any camera pose information or explicit 3D representations during training. The core technical contribution is the Latent Plücker Learner, a component that learns a latent pose space in a self-supervised manner, enabling viewpoint conditioning without ground-truth poses.

Experiments show that UP-LVSM, when trained on a large dataset (66K scenes), achieves state-of-the-art performance, even outperforming methods that require ground-truth pose annotations. The paper argues that this validates their hypothesis and demonstrates that noisy 3D knowledge (such as SfM poses) can be a "performance bottleneck" at scale.

**Strengths:**

1. The paper's core finding—that at scale, it's better to learn 3D from 2D data than to rely on noisy, explicit 3D knowledge (like SfM) —is a significant and impactful claim, which is well-supported by the UP-LVSM results.

2. The analysis in Section 3, which isolates the effects of 3D inductive bias and pose annotation dependence, is a valuable contribution in its own right. The use of dataset subsets to show performance trends (Figures 2, 4, and 5)  is very well done.

3. The proposed UP-LVSM achieves SOTA performance on RealEstate10K, impressively outperforming LVSM (28.82 vs 27.60 PSNR)  despite LVSM having access to ground-truth input poses. This is a very strong result that validates the paper's hypothesis.

4. The Latent Plücker Learner is a novel and well-designed component for learning a latent pose space without supervision. The design thoughtfully considers and addresses the risk of information leakage from the target view.

5. The appendix is not an afterthought but contains crucial, high-quality analysis. Appendix E shows that LVSM are highly sensitive to pose noise, whereas UP-LVSMs are immune. Appendix D solves a key practical limitation of unposed methods.

6. The paper is also exceptionally well written, structured and the figures and tables are all very informative.

**Weaknesses:**

1. The paper's claim to remove "3D inductive biases" and operate without "any 3D knowledge"  is inaccurate. The Plücker ray embedding, which is foundational to the Latent Plücker Learner, is a strong 3D geometric prior. It encodes a line in 3D space. The claim should be more precise, e.g., "without explicit 3D scene representations (like meshes or 3DGS) or camera pose annotations.

2. A major practical drawback of unposed methods is the inability to control the camera for rendering. The paper presents a very simple and effective solution (fine-tuning a linear mapper) in Appendix D. For me personally, this is a crucial insight for making the method practical, yet it is absent from the main paper, potentially leaving readers with the impression that UP-LVSM is not controllable.

**Questions:**

Do you have a sense of the upper bound? For instance, how does UP-LVSM (trained on 66K scenes) compare to a per-scene optimized 3DGS (a different kind of "upper bound")?

In Table 8, the 3D correspondence probing shows UP-LVSM (31.9) performing slightly worse than the off-the-shelf DINOv2 (36.8) at 0-15°. This is counterintuitive, as your model's encoder is fine-tuned on this 3D-aware task. Do you have an explanation for why the specialized model would be worse than the general-purpose one?

---

> ### Author Response · Authors · 2025-11-21
> **Rebuttal for Reviewer n5sM (1/2)**
>
> We thank Reviewer n5sM for the very positive evaluation and for highlighting both the significance of our scalability analysis and the strong performance of UP-LVSM. We appreciate the reviewer’s careful reading and address the concerns and questions below.
>
> - **[W1] Statements on 3D knowledge should be precise.**
>
>   We appreciate the suggestions and have polished the terminology.
>
> - **[W2] Move the camera controllability explanation to the main paper**
>
>   We have moved this section into the main paper.
>
> - **[Q1] Performance upper bound**
>
>   We have provided a detailed explanation below.
>
> - **[Q2] 3D awareness gap between DINOv2 and UP-LVSM**
>
>   We attribute this to key differences in self-supervised learning objectives, as detailed below.
>
> ---
>
> ### **[W1] Statements on 3D knowledge should be precise.**
>
> We appreciate the reviewer’s constructive suggestions and agree that claiming “without any 3D knowledge” could be controversial. It is clear that UP-LVSM bypasses removes explicit 3D scene representations (e.g., meshes, NeRF fields, 3D Gaussians) and all camera pose annotations. However, as the reviewer insightfully points out, the Plücker-inspired latent projection, albeit applied in latent space, potentially introduces a strong geometric prior.
> Following this advice, we have rephrased “without any 3D knowledge” as “with minimal 3D knowledge”, and “remove 3D inductive bias” as “remove explicit scene structures”. We quite agree that these refinements could further improve the precision and the soundness of our work.
>
> ---
>
> ### **[W2] Move the Camera Controllability Explanation to the Main Paper**
>
> We thank the reviewer for highlighting this issue and agree that the linear mapper in Appendix D (real poses → latent plucker) is indeed key to showing that UP-LVSM is controllable at test time. With the additional page allowance, we have moved the controllability discussion into the main paper to make this aspect explicit.

---

> ### Author Response · Authors · 2025-11-21
> **Rebuttal for Reviewer n5sM (2/2)**
>
> ### **[Q1] Performance upper-bound**
>
> First, we try to interpret the reviewer’s question as follows, and hope there is no misunderstanding:
>
> 1. Given dense input views (e.g., more than 300), the 3DGS achieves excellent performance, which somehow defines a “3DGS-quantified upper bound” of the NVS task on a given scene.
> 2. Similarly, UP-LVSM’s performance improves as its training data scales. Given huge data (66K in RealEstate10K), the UP-LVSM performance will also define an upper bound.
> 3. The reviewer asks how the two upper bounds compare.
>
> We believe this is an excellent point to demonstrate the difference between our data-centric feed-forward approach (UP-LVSM) and the classic per-scene optimization approaches (3DGS). To answer it, we include `rebuttal.mp4` in the updated supplementary material, providing a qualitative comparison between UP-LVSM and 3DGS. It suggests:
>
> 1. Under current settings, the 3DGS upper bound (with 318 inputs) is very close to the UP-LVSM upper bound (with only 32 input views, but strong learning priors)
>     - In the first scene of the video, UP-LVSM performs better while 3DGS (318 inputs ver.) struggles with noticeable artifacts and floaters.
>     - In the second scene, UP-LVSM is slightly blurrier but overall comparable.
> 2. However, improving 3DGS’s upper bound requires *capturing even more dense input views* or relying on more carefully designed techniques. In contrast, UP-LVSM can simply benefit from more training data to improve its “upper bound”. As a result, we believe our method is more scalable (and also more useful due to its sparse input compatibility). This contrast is precisely the central value proposition of our work.
>
> We are open to having further discussions if there is anything unclear.
>
> ---
>
> ### **[Q2] 3D Awareness Gap between DINOv2 and UP-LVSM**
>
> We appreciate the insightful observation raised by Reviewer n5sM (as well as Reviewer 6twy) and are happy to have discussions on this point. Here, we would like to share our understanding.
>
> 1. **Common ground.** We share the reviewers’ intuition that, supervised model trained with explicit 3D signals is expected to exhibit stronger 3D awareness than a generic foundation model with self-supervised learning (SSL).
> 2. **Our training objective is closer to SSL than to supervised 3D learning.** UP-LVSM leverages intra-video frame-level consistency as a self-supervisory signal, analogous to how MAE/DINOv2 leverages intra-image patch-level consistency. Neither UP-LVSM nor DINOv2 has access to ground-truth correspondences.
> 3. **However, SSL is not equal to representation learning.** As shown in recent 3D representation learning work (e.g., Sonata [1]), an SSL objective alone does not guarantee strong representations, typically struggling with representation collapse due to “geometric shortcuts”. Effective representation learning relies on specific regularization techniques such as teacher-student distillation and progressive scheduling to obtain strong 3D awareness.
> 4. **Therefore**, given (2) and (3), a probing gap between DINOv2 and UP-LVSM is not too unexpected, especially given that UP-LVSM is not optimized for representation learning and does not employ such regularization.
> 5. **Despite this, we believe closing the gap with techniques like RePA [2] is feasible.** Recently, Efficient-LVSM [3] shows that attaching lightweight representation-alignment heads to LVSM-style encoders can improve probing performance.
> 6. **In conclusion,** it is reasonable that DINOv2 achieves slightly better correspondence recall at very small viewpoint changes (0$^\circ$–15$^\circ$), while UP-LVSM becomes competitive or superior at larger viewpoint changes, where multi-view geometric consistency plays a more dominant role.
>
> We hope this explanation provides a valuable perspective and are willing to have any further discussion.
>
> > *[1] Sonata: Self-Supervised Learning of Reliable Point Representations*
> >
> > *[2] Representation Alignment for Generation: Training Diffusion Transformers Is Easier Than You Think*
> >
> > *[3] Efficient-LVSM: Faster, Cheaper, and Better Large View Synthesis Model via Decoupled Co-Refinement Attention*
>
> ---
>
> ### **Summary**
>
> We thank Reviewer n5sM again for the constructive suggestions, and we believe it is highly valuable to discuss the raised in-depth issues. We hope our clarifications address the reviewer’s concerns, and we welcome further discussion.

---

> > ### Comment · Reviewer_n5sM · 2025-11-26
> > **Answer to rebuttal**
> >
> > Thank you for the detailed rebuttal. After reading my colleagues feedback and the rebuttals, I am still of the opinion that this paper is both timely and insightful and I would encourage acceptance of the paper. My (limited) questions and suggestions have been properly addressed and I believe that with the additional changes/experiments this will be a solid paper for ICLR.

---

> > > ### Author Response · Authors · 2025-11-27
> > >
> > > We appreciate the reviewer's insightful suggestions and constructive feedback, which have significantly enhanced the paper's quality. We also appreciate the valuable discussions regarding the performance upper bound and 3D Awareness, and are happy to confirm that the questions have been well addressed. Again, thanks for your efforts and time during both the review and the discussion period.

---

### Official Review · Reviewer_WSHE · 2025-11-01

**Soundness:** 1
**Presentation:** 3
**Contribution:** 2
**Rating:** 2
**Confidence:** 4

**Summary:**

- The paper analyzes existing Novel View Synthesis (NVS) methods and discovers a core trend: methods that depend less on explicit 3D knowledge (poses, handcrafted 3D representations) benefit more from data scaling and eventually outperform 3D-biased approaches.

- Based on this finding, the authors propose UP-LVSM, a feed-forward, fully data-centric NVS framework that learns implicit 3D structure directly from large-scale 2D images—without camera pose supervision or predefined 3D representations.

- The method introduces a Latent Plücker Learner to infer camera geometry implicitly, enabling state-of-the-art performance in novel view synthesis from sparse, unposed images.

**Strengths:**

- This paper demonstrates that removing explicit 3D priors allows performance to scale significantly with data, outperforming pose-supervised 3D knowledge-driven methods.

- This paper has a pose-free & explicit-3D-free pipeline, a fully feed-forward transformer architecture that works without SfM poses, NeRF/3DGS priors, or handcrafted 3D structures—simplifying training and deployment.

- Extensive experiments show state-of-the-art results and confirm the central hypothesis, providing both conceptual insight and practical contribution to scalable NVS.

**Weaknesses:**

- I have watched the supplementary video. (1) The zoom-in and zoom-out distance is too short. (2) Also, compared to existing models, the method seems to only improve visual quality.

- For datasets where GT is provided, the resolution appears to be very low. I am curious how the method performs on higher-resolution datasets. (If this dataset is the best available choice, I expect the authors to justify that in the rebuttal.)

- The process of simply combining a Transformer with DINOv2 raises concerns regarding novelty. The authors should demonstrate where the novelty lies in this work. What is the novelty contribution of the Latent Plücker Learner?

- The ablation study section is not intuitive. Merely describing components with text makes it difficult to understand what was included or excluded in each ablation setting.

**Questions:**

Mentioned in the weaknesses

---

> ### Author Response · Authors · 2025-11-21
> **Rebuttal for Reviewer WSHE (1/2)**
>
> We thank Reviewer WSHE for the detailed feedback and for highlighting the significance of “providing both conceptual insight and practical contribution to scalable NVS”. We summarize the reviewer’s concerns below:
>
> - **[W1.1] Concerns about zoom distance in the video**
>
>     We would like to justify that this follows the conventional setting.
>
> - **[W1.2] Concerns about the significance of the visual quality improvements**
>
>     We have provided a detailed clarification below to demonstrate the significance.
>
> - **[W2] Concerns about the dataset resolution**
>
>     We would like to clarify that this follows the conventional setting. And it is feasible to obtain a model with a higher resolution.
>
> - **[W3] Concerns about Novelty**
>
>     We have provided a detailed clarification below to highlight our primary novelty.
>
> - **[W4] Improve clarity for ablation studies**
>
>     We appreciate the suggestions and have added an illustration.
>
> ---
>
>
> ### **[W1.1] Concerns about Zoom Distance in the Video**
>
> We thank the reviewer for the comment. The short-motion camera trajectories follow **standard evaluation practice** in feed-forward NVS and strictly match the settings used in NoPoSplat [1], LVSM [2], and other prior works. This convention exists because:
>
> - Typical feed-forward NVS approaches [1,2,3] rely on only *two input views*. Since the task is not generative, these methods must infer novel views solely from visual cues in the inputs.
> - Consequently, the two input views must overlap. Otherwise, rendering large unseen regions is inherently impractical, as discussed in Appendix G.
> - As a result, the current camera moving distance is already challenging in many cases. As shown in the supplementary video (1:36-1:46), the model has to infer 3D correspondence from two little overlapped inputs.
>
> Despite this, we agree that longer trajectories would better showcase applicability and scalability. It is easy to extend our 2-view model for longer trajectory rendering by concatenating. Given a long camera trajectories $V\_1,V\_2,\dots$, we simply render short transitions $V\_1\rightarrow V\_2,V\_2\rightarrow V\_3,\dots$ and concatenate them. We have updated an example `rebuttal.mp4` in the supplementary material, where our method achieves good NVS performance under long camera trajectories.
>
> We hope these additions address the reviewer’s concern.
>
> > *[1] No Pose, No Problem: Surprisingly Simple 3D Gaussian Splats from Sparse Unposed Images*
> >
> > *[2] LVSM: A Large View Synthesis Model with Minimal 3D Inductive Bias*
> >
> > *[3] pixelSplat: 3D Gaussian Splats from Image Pairs for Scalable Generalizable 3D Reconstruction*
>
> ---
>
> ### **[W1.2] Concerns about Significance of the Visual Quality Improvements**
>
> We appreciate the reviewer’s concern regarding the perceived magnitude of improvement in the supplementary video. We would like to clarify a few important points to better contextualize the significance of our results:
>
> 1. **Our method is completely pose-free,** without access to camera poses during either training or inference. In contrast, baselines such as NoPoSplat require ground-truth or COLMAP-estimated target poses. Achieving superior rendering quality while removing all pose supervision is a substantial capability improvement.
> 2. **2D visual quality is the primary indicator of 3D reasoning in NVS**. Incorrect or inconsistent 3D inference leads to blurred, distorted, or collapsed renderings. As shown in the first and second rows of Fig. 7, UP-LVSM produces noticeably more stable and detailed results than NoPoSplat, indicating stronger 2D–3D understanding despite having *less* 3D information.
>
> We hope this clarifies why these visual quality improvements are meaningful.
>
> ---
>
> ### **[W2] Concerns about the Dataset Resolution**
>
> We thank the reviewer for the comment. Similar to W1.1, the low resolution (typically 256x256 or 224x224) is a widely used setting in prior works [1,2,3] that we also followed. As directly training feed-forward NVS models at high resolution leads to both instability and inefficiency, starting with a low resolution is important for large transformer-based models like LVSM and our UP-LVSM for stable and efficient training. This is the primary reason that existing works typically standardize evaluation at low resolution.
>
> However, after training and evaluating at 224x224, we can naturally extend our method to higher resolution (e.g., 518x518) via fine-tuning, as described in the last paragraph of Appendix I.1. Also, the `rebuttal.mp4` video in the updated supplementary material is synthesized at 518x518, serving as an example. Further high-resolution fine-tuning is certainly feasible, and we believe our framework is generalizable.
>
> > *[1] No Pose, No Problem: Surprisingly Simple 3D Gaussian Splats from Sparse Unposed Images*
> >
> > *[2] LVSM: A Large View Synthesis Model with Minimal 3D Inductive Bias*
> >
> > *[3] pixelSplat: 3D Gaussian Splats from Image Pairs for Scalable Generalizable 3D Reconstruction*

---

> ### Author Response · Authors · 2025-11-21
> **Rebuttal for Reviewer WSHE (2/2)**
>
> ### **[W3] Concerns about Novelty**
>
> We thank the reviewer for raising this question. While our model uses a DINOv2 tokenizer together with a transformer-based architecture—which may appear conventional—the core novelty does not lie in inventing a single isolated module, but in the high-level design of **a fully data-centric NVS framework**, including the architectural formulation, training objective, and data strategy. Our contributions focus on **how to enable fully feed-forward, pose-free NVS at scale**. Specifically, we provide:
>
> 1. **The analysis: the less you depend, the more you learn.** We provide a systematic analysis showing that heavy reliance on explicit 3D inductive bias or SfM pose annotations fundamentally limits scalability. This finding is meaningful, not only indicating that pure Transformer-based methods would surpass their bias-driven counterparts at large data scale, but also demonstrating the counterintuitive conclusion that camera poses could serve as indirect 3D bias and become a bottleneck.
> 2. **A scalable, pose-free feed-forward NVS framework.** Building on this analysis, we propose a data-centric feed-forward NVS framework that benefits greatly from scalability, achieving state-of-the-art NVS performance even without any explicit 3D supervision. This reveals the potential of scaling feed-forward NVS with much larger-scale video data.
> 3. **Latent Plücker Learner as a practical technique to reduce pose dependence.** In the context of (1) and (2), our Latent Plücker Learner serves as a practical implementation of the “the less you depend” principle to bypass the explicit pose dependence. Its novelty lies in incorporating the Plücker embedding with the bottleneck architecture, which enables the model to learn meaningful pose information in a self-supervised manner while avoiding noticeable information leakage.
>
> Among these, the analysis and the resulting framework are the primary contributions. Our design of the specific module (Latent Plücker Learner) is also crucial, yet not the central contribution. We hope this clarifies our central contributions and are happy to provide further clarification or have further discussions.
>
> ---
>
> ### **[W4] Improve Clarity for Ablation Studies**
>
> We thank the reviewer for the valuable suggestion and agree that the textual explanation of the ablation study could be made clearer. With the additional page allowance during rebuttal, we have updated it with a detailed explanation and an illustration in Section 4.3. We believe the reviewer's suggestion effectively improves the readability.
>
> ---
>
> ### **Summary**
>
> We thank Reviewer WSHE again for the thoughtful comments. We hope our clarifications better convey both the conceptual and practical contributions of our work, and we welcome further discussion.

---

> > ### Comment · Reviewer_WSHE · 2025-11-26
> >
> > The file rebuttal.mp4 is still difficult to understand. Could you please clearly specify which parts correspond to your actual outputs?
> >
> > You also mention in Appendix I.1 that the resolution can be upscaled, but this point should be explicitly stated and clearly demonstrated here as well.
> >
> > I would be willing to consider increasing the rating once these issues are properly addressed.

---

> > > ### Author Response · Authors · 2025-11-27
> > >
> > > Thanks for raising this point. We are glad to follow the reviewer's helpful suggestion to improve the clarity of `rebuttal.mp4` by providing specific contextualization regarding the Novel View Synthesis (NVS) protocols.
> > >
> > > We have **updated the video** to include textual indicators and comparisons. Below, we provide a detailed explanation of the problem setting, our specific inference protocol, and the resolution upscaling demonstration.
> > >
> > > ---
> > >
> > > ### **1. Clarification on Long Sequence Rendering**
> > >
> > > To clarify exactly which parts of the video correspond to actual outputs, we first outline the standard evaluation protocol used in this field, followed by our specific implementation in the video.
> > >
> > > **A. Context: The Video Reconstruction Benchmark**
> > > In the context of NVS, "video reconstruction" is a standard benchmark used to evaluate a model's ability to infer 3D structure:
> > > * **The Task:** Given a video of length $N$, a subset of $N\_{\text{in}}$ frames is selected as "input views" (seen). The model must utilize visual cues from these inputs to synthesize the remaining $N - N\_{\text{in}}$ frames (unseen).
> > > * **The Goal:** The model reconstructs the entire video sequence. The fidelity of the reconstructed frames (particularly the unseen ones) serves as the primary metric for the model's understanding of the underlying 3D scene.
> > >
> > > **B. Comparison: 3DGS vs. UP-LVSM**
> > > * **Traditional Approach (e.g., 3D Gaussian Splatting):** Standard methods typically take all $N\_{\text{in}}$ frames at once, estimate camera poses for the whole sequence (using tools like COLMAP), and optimize a global scene representation (e.g., a cloud of 3D Gaussians) to reconstruct the video.
> > > * **Our Approach (UP-LVSM):** In contrast, UP-LVSM is a feed-forward, two-view method. We do not optimize a global scene. Instead, we perform piecewise inference. For a long trajectory, we take two inputs (e.g., $V\_A, V\_B$), render the transition between them, and repeat this process for subsequent pairs ($V\_B, V\_C$), concatenating the results.
> > >
> > > **C. Specific Configuration in the Video**
> > > The video demonstrates the "Coffee Bar" scene (318 frames total).
> > > 1.  **Input Sampling:** We uniformly sample 32 views as inputs (roughly every 10 frames: $V\_0, V\_{10}, V_{20}, \dots, V_{317}$).
> > > 2.  **Inference:**
> > >     * We input pair $\{V\_0, V\_{10}\}$ to generate the segment $V\_{0 \to 10}$.
> > >     * We input pair $\{V\_{10}, V\_{20}\}$ to generate the segment $V\_{10 \to 20}$.
> > >     * This continues until the full trajectory is stitched together.
> > > 3.  **The Output:** Consequently, **every frame shown in the video is a rendered output synthesized by UP-LVSM.** The model has "seen" the specific input frames (approx. 1 in every 10), but all intermediate frames are inferred novel views.
> > >
> > > **D. Visual Updates**
> > > To assist the viewer, the updated `rebuttal.mp4` now includes:
> > > * **Real-time Frame Index:** Tracking the position in the sequence.
> > > * **Input Indicators:** Explicit text labels indicating whether specific frames are provided as inputs to the model.
> > >
> > > ---
> > >
> > > ### **2. Demonstration of Upscaled Resolution**
> > >
> > > The reviewer correctly noted that while we mentioned resolution upscaling in Appendix I.1, it was not explicitly stated and not clearly demonstrated. We have addressed this in the updated `rebuttal.mp4` .
> > >
> > > * **Explicit Statement:** We explicitly state the resolution settings.
> > > * **Clear Demonstration:** A side-by-side comparison of the original rendering ($224 \times 224$ resolution) versus the fine-tuned version ($518 \times 518$ resolution) is added.
> > >
> > > The updated video results explicitly demonstrate that UP-LVSM can scale effectively to higher resolutions for better visual quality. Notably, this improvement simply requires higher-resolution training data instead of specialized architectural modifications, highlighting the scalable value of our UP-LVSM.
> > >
> > > ---
> > >
> > > We hope these explanations and the updated visualizations will effectively address the reviewer's concerns. We are grateful for the constructive feedback, which has significantly improved the presentation of our work.

---

### Official Review · Reviewer_6twy · 2025-11-03

**Soundness:** 4
**Presentation:** 4
**Contribution:** 3
**Rating:** 8
**Confidence:** 4

**Summary:**

The authors posit the principle “the less you depend, the more you learn”—arguing that reducing reliance on explicit 3D priors (like NeRF/3DGS representations or SfM-derived camera poses) improves scalability and generalization as training data scales.

This message extends upon LVSM and removes the training- and test- time pose requirements to make the framework even more data centric. The authors introduce a Latent Plücker Learner to infer camera poses in a self-supervised manner.

The paper presents empirical analysis across datasets (RealEstate10K, DL3DV, ACID, Objaverse) and compares UP-LVSM to bias-driven and pose-dependent methods (MVSplat, NoPoSplat, LVSM). Results show that UP-LVSM scales better with data and even surpasses 3D-supervised models in rendering fidelity and generalization.

**Strengths:**

- The paper is easy to follow and the authors promise to release code for reproducibility.

- The paper evaluates across diverse datasets and metrics, providing ablations, scalability curves, and qualitative examples. The performance gains are consistent and very decent (e.g., +1 PSNR improvements over LVSM on large-scale data).

- Removing the requirement of training- and test- time pose annotations have huge potential of scaling to much larger-scale datasets; hence this work (along with RayZer) opens up a lot of new possibilities.

**Weaknesses:**

- UP-LVSM seems to have separately trained models on the RealEstate10k scene data and the Objaverse object data. It remains unclear to me if the latent plucker learning component can be trained on a mix of scene and Objaverse datasets. From the scalability perspective, it makes sense to have a method capable of ingesting all available data sources.

- Tab. 4 is presented in a confusing way; it's unclear what baseline the performance gain is evaluated against. Moreover, I think it makes more sense to also include the absolute PSNR/SSIM/LPIPS values, rather than just providing the relative changes.

**Questions:**

- in Tab. 8, it seems to me that DINOv2, though not explicitly trained on 3D tasks, seem to provide more accurate correspondence estimation than UP-LVSM which is trained on the 3D NVS task. I guess I find it a bit hard to understand.

- in Tab. 5 and Fig. 7, how is the target pose provided to different methods? Did all the methods use the same ground-truth target pose? Or they consume poses estimated by UP-LVSM?

- Line 1105 mentions that "However, for the experiments reported in the main paper, we standardize all evaluations to the 224 ×224 setting, including all baseline comparisons." I wonder if this is fair to baselines if their provided checkpoint was trained only on 256x256.

---

> ### Author Response · Authors · 2025-11-21
> **Rebuttal for Reviewer 6twy (1/2)**
>
> We sincerely thank Reviewer 6twy for the thoughtful and constructive feedback, which we believe will substantially strengthen our work. Below, we address the reviewer’s remaining concerns in detail and have incorporated revisions accordingly.
>
> - **[W1] Mixed training on scene-level and object-level data**
>
>     Yes, it is feasible. We have validated this.
>
> - **[W2] Improve Table 4 Representation**
>
>     Thanks. We have reorganized Table 4.
>
> - **[Q1] Why DINOv2 sometimes surpasses UP-LVSM in 3D awareness metrics (Table 8)**
>
>     We attribute this to fundamental differences in self-supervised learning objectives, which we will detail below.
>
> - **[Q2] Target pose source in Table 5 and Fig. 7**
>
>     Only UP-LVSM uses its predicted target poses, while other methods use target poses from the “ground truth” COLMAP annotations provided by the dataset.
>
> - **[Q3] Fairness of the 224×224 evaluation.**
>
>     We train all baselines from scratch at 224×224, instead of using their official checkpoints trained at 256x256.
>
>
> ---
>
> ### **[W1] Mixed Training on Scene-level and Object-level Data**
>
> We highly agree with the reviewer that ingesting all available data sources is key to scalability.
>
> Our decision (the same as other approaches [1,2,3]) to train separate models is driven by the substantial distribution gap between camera trajectories in RealEstate10K (forward-facing, sequential) and Objaverse (object-centric, unordered). Consequently, mixed training often leads to instability.
>
> Existing approaches [1,2] typically address this by training separate models with different hyperparameters. For example, LVSM [1] trains a scene-level model under 2 input views and a perceptual loss weight of 0.5, while requiring 4 input views and a perceptual loss weight of 1.0 for the object-level model (its Section 4.2). Rayzer [2] applies a special pose interpolation on Objaverse (its Appendix A). Other works like CUT3R [4] manage to conduct a mixed training process, at the cost of a relatively complex training recipe to ensure stability.
>
> To address the reviewer’s concern and to verify the scalability of our Latent Plücker Learner, we conducted supplementary mixed-data experiments combining RealEstate10K and Objaverse (a subset due to the limited rebuttal period). By re-rendering our Objaverse data following a sequential camera trajectory instead of an unordered set of sparse cameras, we found it possible to train on mixed data at the cost of a slight degradation (0.58dB in PSNR), which might require progressive training strategies to mitigate. We have added a discussion of this experiment in the revised version (Appendix I.3).
>
> > *[1] LVSM: A Large View Synthesis Model with Minimal 3D Inductive Bias*
> >
> > *[2] RayZer: A Self-supervised Large View Synthesis Model*
> >
> > *[3] GS-LRM: Large Reconstruction Model for 3D Gaussian Splatting*
> >
> > *[4] Continuous 3D Perception Model with Persistent State*

---

> ### Author Response · Authors · 2025-11-21
> **Rebuttal for Reviewer 6twy (2/2)**
>
> ### **[W2] Improve Table 4 Representation**
>
> We sincerely appreciate the reviewer’s valuable suggestion. The original table is designed to focus on relative gains to highlight scalability trends due to strict page limits. With the additional page allowance during rebuttal, we have revised Table 4 to also include absolute metrics, matching the detail level of Table 3.
>
> We agree that this could be much more informative and provide a clearer picture of how UP-LVSM compares to baselines across different data scales.
>
> > For example, on real-world datasets, UP-LVSM consistently holds better scalability than LVSM, while the trend is inverted on the synthetic Objaverse dataset, where absolutely accurate poses are provided (as discussed in Appendix E, Table 13).
>
> ---
>
> ### **[Q1] 3D Awareness Gap between DINOv2 and UP-LVSM**
>
> We appreciate the insightful observation raised by Reviewer 6twy (as well as Reviewer n5sM) regarding the probing gap of 3D awareness. We would like to offer a detailed explanation to share our understanding.
>
> 1. **Common ground.** We share the reviewers’ intuition that, supervised model trained with explicit 3D signals is expected to exhibit stronger 3D awareness than a generic foundation model with self-supervised learning (SSL).
> 2. **Our training objective is closer to SSL than to supervised 3D learning.** UP-LVSM leverages intra-video frame-level consistency as a self-supervisory signal, analogous to how MAE/DINOv2 leverages intra-image patch-level consistency. Neither UP-LVSM nor DINOv2 has access to ground-truth correspondences.
> 3. **However, SSL is not equal to representation learning.** As shown in recent 3D representation learning work (e.g., Sonata [1]), an SSL objective alone does not guarantee strong representations, typically struggling with representation collapse due to “geometric shortcuts”. Effective representation learning relies on specific regularization techniques such as teacher-student distillation and progressive scheduling to obtain strong 3D awareness.
> 4. **Therefore**, given (2) and (3), a probing gap between DINOv2 and UP-LVSM is not too unexpected, especially given that UP-LVSM is not optimized for representation learning and does not employ such regularization.
> 5. **Despite this, we believe closing the gap with techniques like RePA [2] is feasible.** Recently, Efficient-LVSM [3] shows that attaching lightweight representation-alignment heads to LVSM-style encoders can improve probing performance.
> 6. **In conclusion,** it is reasonable that DINOv2 achieves slightly better correspondence recall at very small viewpoint changes (0$^\circ$–15$^\circ$), while UP-LVSM becomes competitive or superior at larger viewpoint changes, where multi-view geometric consistency plays a more dominant role.
>
> We hope this provides a valuable perspective and are willing to have any further discussion.
>
> > *[1] Sonata: Self-Supervised Learning of Reliable Point Representations*
> >
> > *[2] Representation Alignment for Generation: Training Diffusion Transformers Is Easier Than You Think*
> >
> > *[3] Efficient-LVSM: Faster, Cheaper, and Better Large View Synthesis Model via Decoupled Co-Refinement Attention*
>
> ---
>
> ### **[Q2] Target Pose Source in Table 5 and Fig. 7**
>
> For all experiments in Table 5 and Figure 7, only UP-LVSM relies on its predicted poses, while any other baseline (e.g., MVSplat, NoPoSplat, LVSM, PT-LVSM) uses target poses taken from the datasets’ own COLMAP annotations.
>
> We intentionally design this comparison to highlight that despite the baselines having access to these oracle poses, our UP-LVSM still achieves superior or competitive rendering fidelity.
>
> We have added a clear statement in Section 4.3 to avoid ambiguity.
>
> ---
>
> ### **[Q3] Fairness of the 224×224 evaluation**
>
> We fully agree that evaluating a model trained at 256×256 on 224×224 inputs would be unfair. To ensure strict fairness, we do not use any official checkpoints.
>
> Instead, we retrain all baseline methods from scratch—this retraining is also necessary for the scalability experiments in Section 3, where the amount of training data varies—using the same 224×224 resolution with a patch size of 14 and training split as UP-LVSM. All training follows the official implementations of each baseline.
>
> We have added an explicit statement in Section 4.3.
>
> ---
>
> ### **Summary**
>
> We thank Reviewer 6twy again for the insightful comments, which have helped us significantly improve the clarity and completeness of the manuscript. We are highly willing to have any further clarification.

---

### Author Response · Authors · 2025-11-21
**Revision Summary**

*Dear Reviewers,*

We are greatly encouraged by the supportive assessments of ***Reviewers 6twy*** and ***Reviewer n5sM*** on the value of our work. And we sincerely appreciate the constructive feedbacks from ***Reviewer WSHE*** and the detailed suggestions from ***Reviewer KQEM***. Below, we outline the main revisions addressing the remaining concerns raised by ***Reviewers 6twy, WSHE, n5sM, and KQEM***. All updates in the revised manuscript are highlighted for clarity.

---

### **Main Paper & Appendix**

- We have updated Table 4 to include absolute metrics, matching the detail level of Table 3 ***(Reviewer 6twy, W2)***.
- We have detailed experimental setup in Section 4.3 ***(Reviewer 6twy, Q3)***.
- We have detailed ablation studies and have added an illustration in Section 4.3 ***(Reviewer WSHE, W4)***.
- We have polished terminology like *“3D inductive biases”* or *“any 3D knowledge”* to make our claims more precise ***(Reviewer n5sM, W1)***.
- We have moved camera controllability explanation into Section 4.4 of the main paper ***(Reviewer n5sM, W2)***.
- We have included experiments on mixed object-level and scene-level training in Appendix I ***(Reviewer 6twy, Q1)***.

> Note: For consistency, new tables and figures added during rebuttal are labeled using A/B/C prefixes. And the original controllability appendix section is retained temporarily for label consistency and will be removed after rebuttal.

---

### **Supplementary Materials**

- We include an original version of our paper in the supplementary materials.
- We add a new video results ( `rebuttal.mp4` ) in the supplementary materials, demonstrating long distance rendering ***(Reviewer WSHE, W1 & W2; Reviewer KQEM, W2)*** and comparisons with 3DGS ***(Reviewer n5sM, Q1)***.

---

We sincerely appreciate the reviewers’ time and constructive feedback, which have substantially improved the clarity and quality of our manuscript.

*Best,*

*Authors*

---

### Meta-Review · Area_Chair_2QkZ · 2026-01-06

**Summary:**

The paper studies the data scalability of 3D models with or without 3D inductive bias. Two inductive biases are considered: the explicit scene representations, and the explicit camera models. The paper illustrated that the data scalability is better with less inductive bias. The data scalability is measured by the performance gain on homogeneous data. Lastly, the paper proposed a method to do unsupervised reconstruction training (i.e., removing both the inductive biases of explicit scene representation and the camera models).

I think that overall it's a good paper, no severe concerns are raised by reviewers. The reviewer requested clarity on experimental setups, and some more results. These are provided in rebuttal. The remaining concerns are either beyond scope of the paper or minor.

**Reviewer Concerns:**

Remaining concerns:

1. **Effectiveness of DINOv2 feature** (6twy). The author explains the possibility in text without experimental proof. However, this is a minor point.

2. **Effectiveness on high resolution** (WSHE, 6twy). 6twy found that the paper's experimental resolution is lower than other papers (224 vs 256). WSHE wonders the effectiveness on high resolution. Both concerns are partially resolved. The rebuttal claimed that all baselines are retrained with lower resolution to match their experimental setup. The rebuttal also included an updated video to visualize the higher resolution results (with 518x518 after fine-tuning). However, no quantitative results are provided for high resolution. In my opinion, this concern is minor. Fine-tuning with 518x518 and showing the result would be an convincing experiment but it does not fully hurt the validity of the paper. Also, we would not force the author to train high-res model from scratch as it can take too much compute.

7. **failure modes are under-explored** (KQEM). The rebuttal provides some example but lacks systematical analysis. I think that this point is valid. As the failure mode can be different between explicit and implicit method, fully relying on visual metrics built on explicit method might not reliable. However, I do think that this is beyond the scope of the paper. I encourage some relevant discussions in paper but would not enforce a solution.

Resolved concerns:

1. **Missing mixed object + scene training results** (6twy)  Solved during rebuttal with results.

2. **Novelty** (WSHE); seems that the reviewer does not hold this concern in the response to the rebuttal.

3. **Unclear description of ablation experimental setup** (WSHE); Solved in rebuttal with update paper.

4. **Statements on 3D knowledge should be precise.** (n5sM); Solved in rebuttal.

5. **Camera Controllability** (n5sM); sounds like a misunderstanding and is resolved.

6. **performance in low-data regimes** (KQEM); more like a writing issue.

**Reviewer Scores:**

The original score is 2 6 8 8. After rebuttal, it might be **4 6 8 8**.

Reviewer 6twy: original score of 2. Main concerns are resolved in rebuttal and mentions the willing to improve the score.

Reviewer KQEM: original score of 6. The review has some redundancy and I would lower the confidence of this review score.

---

### Decision · Program_Chairs · 2026-01-26

Accept (Poster)